



# Using satellite measurements and mesoscale modelling to understand the contribution to an extreme air pollution event in India

Ashique Vellalassery[1], Dhanyalekshmi Pillai, [1,*], Julia Marshall [2,#], Christoph Gerbig[2], Michael Buchwitz[3], and Oliver Schneising[3].

[1] Indian Institute of Science Education and Research Bhopal (IISERB), Bhopal, India

[*] also at Max Planck Partner Group (IISERB) affiliated with the Max Planck Society Munich, Germany

[2] Max Planck Institute for Biogeochemistry, Jena, Germany

[#] now at Deutsches Zentrum für Luft- und Raumfahrt, Institut für Physik der Atmosphäre, Oberpfaffenhofen, Germany

[3] Institute of Environmental Physics (IUP), University of Bremen, Bremen, Germany

*Correspondence to:* Dhanyalekshmi Pillai (*dhanya@iiserb.ac.in*, kdhanya@bgc-jena.mpg.de)





**Abstract**
Several ambient air quality records corroborate severe and persistent degradation of air quality
over North India during the winter months with evidence of a continued increasing trend of
pollution across the Indo-Gangetic Plain (IGP) over the past decade. A combination of
atmospheric dynamics and uncertain emissions, including the post-monsoon agricultural
stubble burning, make it challenging to resolve the role of each individual factor. Here we
demonstrate the potential use of an atmospheric transport model, the Weather Research and
Forecasting model coupled with chemistry (WRF-Chem) to identify and quantify the role of
transport mechanisms and emissions on the occurrence of the pollution events. The
investigation is based on the use of CO observations from TROPOspheric Monitoring
Instrument (TROPOMI), onboard the Sentinel 5-Precursor satellite, and the surface
measurement network as well as WRF-Chem simulations to investigate the factors contributing
to CO enhancement over India during November 2018. We show that the simulated column-
averaged dry air mole fraction (XCO) is largely consistent with TROPOMI observations with a
spatial correlation coefficient of 0.87. The surface-level CO concentrations show larger
sensitivities to boundary layer dynamics, wind speed, and diverging source regions, leading to
a complex concentration pattern and reducing the observation-model agreement with a
correlation coefficient ranging from 0.41 to 0.60 for measurement locations across the IGP. We
find that daily satellite observations can provide a first-order inference of the CO transport
pathways during the enhanced burning period, and this transport pattern is reproduced well in
the model. By using the observations and employing the model at a comparable resolution, we
confirm the significant role of atmospheric dynamics as well as residential, industrial and
commercial emissions in the production of the exorbitant level of air pollutants in North India.
We find that biomass burning plays only a minimal role in both column and surface
enhancements of CO, except for in the state of Punjab during the high pollution episodes.
While the model reproduces observations reasonably well, a better understanding of the factors
controlling the model uncertainties is essential to relate the observed concentrations to the
underlying emissions. Overall, our study emphasizes the importance of undertaking rigorous
policy measures, mainly focusing on reducing residential, commercial and industrial emissions
in addition to actions already underway in the agricultural sectors.



## 1. Introduction

Biomass burning (BB) has been recognized as the second-largest source of radiatively and chemically active trace gases (e.g. CO, $CO_2$, and $SO_2$) and aerosols (e.g. PM10, and PM2.5) in the global atmosphere, which has significant implications for climatic change and human health (Andreae, 2001; Bond, 2004; Crutzen and Andreae, 1990; Guenther et al., 2006; Kaiser et al., 2012; van der Werf et al., 2017). According to previous reports, BB alone accounts for 59% of Black Carbon (BC) emissions, one-third to one-half of global carbon monoxide (CO) and 20% of nitrogen oxide (NOx) emissions (Akagi et al., 2011; Andreae and Jolla, 2019). Based on the model estimates of Ward et al., (2012), in the absence of fire-related emissions, there would be a reduction of about 40 ppm $CO_2$ from the current atmospheric concentration level, indicating the importance of fire activities for the global carbon budget.

In India, emissions from open-biomass burning include significant contributions from agricultural crop residue burning in addition to forest and grassland fires and play an essential role in terms of releasing total carbon content to the atmosphere. Agricultural stubble burning during the post-harvesting period is one of the main kinds of biomass burning practices used in India to clear the land to make it suitable for the next crop (Tai-Yi, 2012; Zha et al., 2013). According to previous estimates, crop waste open burning, which includes its use in residential heating and cooking, is responsible for 78-83% (116–289 Tg yr$^{-1}$) of the total biomass burned in India during the year 2001 while rest of the contributions are from forest fires (Venkataraman et al., 2006). As per the previous studies, the primary crop residues generated in India are rice straw (112 Mt), wheat straw (109.9 Mt), rice husk (22.4 Mt), sugarcane tops (97.8 Mt) and bagasse (101.3 Mt), the major part of which is burnt in the open air (Lasko and Vadrevu, 2018). Most of these burning activities are found over the northern part of India along the foothills of the Himalayas known as the Indo-Gangetic Plains (hereafter called the IGP). The IGP is a highly populated and very important agro-eco-region in South-Asia, which includes the states of Punjab, Haryana, Bihar, Uttar Pradesh and West Bengal. The region occupies nearly 20% of the total geographical area of India and contributes about 42% to India's total food grains production (Tripathi et al., 2007). Based on VIIRS (Visible Infrared Imaging Radiometer Suite) thermal anomalies, a recent study has estimated burnt crop residues of 20.4 Mt and 9.6 Mt in Punjab and Haryana for the agricultural year 2017-18 in which most of the residue burnt (>90%) at the field was from rice and wheat crops (Singh et al., 2020).

Episodes of pollution events are a major concern in the IGP region, which worsen during post-monsoon and winter seasons (Cusworth et al., 2018; Dekker et al., 2019; Girach and Nair, 2014). According to the World Air Quality Report 2019 based on ambient PM2.5 concentration, fourteen of the top twenty most polluted cities in the world are located in the IGP region, which also includes India's capital region, Delhi. Earlier studies and reports attributed this to several reasons, mainly crop residue burning over Punjab and Haryana, the two adjoining states of India's capital city Delhi (Girach and Nair, 2014; Gupta et al., 2004; Sidhu et al., 1998). However, the contributions from different source sectors and source regions on Delhi's pollution levels still remain highly uncertain, which hinders the implementation of definitive measures to address pollution events. A recent study reported a general lack of reliable data and research efforts on biomass burning related issues on environment and human



health (Yadav et al., 2018). Since agricultural stubble burning is a practice prohibited by law in
India, official surveys conducted to estimate the extent of fire emission are not reliable. There
is, therefore, a critical need to improve the current knowledge base to help to make future
policies and implement mitigation strategies.
Kaiser et al., (2012) demonstrated an approach for calculating biomass-burning emissions by
assimilating satellite-based fire radiative power (FRP) observations in which the combustion
rate and trace gas emissions are subsequently derived with land/cover-specific conversion
factors and emission factors compiled through literature surveys. While the FRP-based
approach has clear advantage in enhancing accuracy compared to other inventory/based
datasets such as the Global Fire Emission Database (GFED), several studies have indicated
inaccuracies in the derived biomass burning products due to instrument limitations and usage
of conversion factors (Cusworth et al., 2018; Dekker et al., 2019; Huijnen et al., 2016; Kaiser
et al., 2012; Mota and Wooster, 2018). The recent availability of greenhouse gas satellite
observations with unprecedented data density at high spatial and temporal resolution paves the
more direct way for a detailed study on the origin, distribution and extent of trace gas levels
over a vast domain on a monthly to daily basis. Carbon monoxide (CO) is one of the major
gases emitted from biomass burning and incomplete fossil fuel combustion. The major sink of
CO is reaction with the hydroxyl radical (OH) to form $CO_2$ and precursor tropospheric ozone.
The lifetime of CO in the atmosphere is between several weeks and several months and varies
with the location and season depending on the oxidizing capacity of the environment (Jaffe,
1968). Compared to $CO_2$ and $CH_4$, the short lifetime of CO makes it easier to detect from the
background concentration level and thus it can be a good tracer of pollution transport (Dekker
et al., 2017). Therefore, CO can be used as a proxy for the anthropogenic emissions of other
pollutants, for example, emissions of important GHGs such as carbon dioxide (Gamnitzer et
al., 2006).
The TROPOspheric Monitoring Instrument (TROPOMI), onboard the Sentinel 5-Precursor
satellite, has been measuring various trace gases, including CO since November 2017
(Landgraf et al., 2016; Borsdorff et al., 2018a, 2019b; Schneising et al., 2019, 2020).
TROPOMI measures with high spatial (7 km × 7 km) and temporal resolution (global daily
coverage, not accounting for cloud and aerosol contamination). The unprecedented data
density, with high spatial and temporal resolution, makes TROPOMI useful for getting
information from city-scale to large-scale. The validation of the TROPOMI retrieval with
ground-level measurements and model simulations has confirmed the high quality of the
measurements, with a high signal to noise ratio, indicating the usefulness of the data collected
(Borsdorff, 2018a, 2018b; Schneising et al., 2019, 2020).
In this study, we make use of carbon monoxide (CO) observations from TROPOMI (see Sect.
2.1) and the surface measurement network to investigate different regional sources of CO in
terms of their contribution to the total column and surface-level concentrations during high
pollution episodes in the winter season. By comparing CO measurements with high-resolution
model simulations generated by WRF-Chem-GHG, we aim to understand the contribution of
different sources to the observed CO enhancement. In particular, we focus on CO enhancement
caused by the emissions from both biomass burning and anthropogenic activities and their





relative roles in the severe air pollution of major cities nearby. This paper aims to address the
following questions: 1) How large is the CO enhancement over northern India detected by
TROPOMI during the agricultural stubble burning period? 2) What is the regional contribution
of CO emissions over India during the entire year 2018? 3) How good is the agreement
between the WRF-Chem-GHG and the observations both at ground level and integrated across
the column? 4) How does the column respond to the spatio-temporal variations of surface
emissions, particularly biomass emissions? and 5) What is the role of different emission
sources in terms of their contribution to the enhanced concentration level during the high
pollution episodes over India? An analysis focusing on identifying the sources contributing to
the high pollution event in North India during November 2017 using WRF modelling and
TROPOMI preliminary operational data was reported in Dekker et al., 2019, but here we
present the analysis for the succeeding year, i.e. November 2018, which also differs from the
previous study as follows: the present study (1) uses the retrievals from both WFM-DOAS
(Schneising et al., 2019, see Sect. 2.1) and TROPOMI/SICOR algorithms (Landgraf et al.,
2016) (2) examines the regional distribution of CO for the entire year, (3) employs different
model configuration such as model domain size, vertical eta levels, and planetary boundary
layer scheme, (4) prescribes a different anthropogenic emission inventory that also includes
hourly variations, and (5) utilizes the entire month, which includes biomass burning and non-
biomass burning periods to get a more detailed view of the dispersion to nearby places.
**2. Data**
**2.1. TROPOMI column observations**
The TROPOspheric Monitoring Instrument (TROPOMI), onboard the Sentinel 5-Precursor
satellite (S5P), has been measuring various trace gases, including CO since November 2017
(Landgraf et al., 2016; Borsdorff et al., 2018a, 2018b; Schneising et al., 2019, 2020). The
TROPOMI instrument consists of a shortwave infrared nadir viewing imaging spectrometer,
which measures radiances around 2.3 μm wavelength, from which the total column mixing
ratio (XCO) is retrieved (Schneising et al., 2019; Landgraf et al., 2016). Due to the wide swath
of about 2600 km, the instrument is able to cover the whole globe on a daily basis, capturing
full scenes of continuous observations in cloud-free conditions (Schneising et al., 2019, 2020).
As a result of the observation of reflected solar radiation in the SWIR part of the solar
spectrum, TROPOMI yields atmospheric carbon monoxide measurements with high sensitivity
to all altitude levels including the planetary boundary layer and is thus well-suited to study
emissions from fires (Schneising et al., 2020).
For this study, we use TROPOMI CO data for November 2018 retrieved using the scientific
algorithm, the Weighting Function Modified Differential Optical Absorption Spectroscopy
(WFM-DOAS) optimised to retrieve vertical columns of carbon monoxide and methane
simultaneously (Schneising et al., 2019). Additionally, we use TROPOMI operational data
(TROPOMI/SICOR CO, Borsdorff et al., 2018a, 2019b) to examine the consistency of these
two observational products over India. The SICOR and WFMD-DOAS algorithms differ in
many aspects including radiative transfer models, inversion schemes and the quality filtering





method used. Whereas WFMD retrievals are limited to cloud free scenes, SICOR aims to retrieve CO columns for cloudy ground pixels also. A global comparison between these two datasets from December 2018 (Schneising et al., 2019) shows a very similar spatial CO pattern for both algorithms with a high correlation coefficient of 0.98 and a regression factor close to the 1:1 line, confirming good agreement between the two datasets. An overview of the TROPOMI datasets used in this study is provided in Table 1 and additional details are provided in the following two sub-sections.

### 2.1.1. Scientific TROPOMI WFMD CO product

The WFM-DOAS retrieval algorithm was initially developed for the SCIAMACHY instrument onboard the ENVISAT satellite (Buchwitz et al., 2006, 2007; Schneising et al., 2011, 2014) and has recently been adjusted for XCO retrieval from TROPOMI (Schneising et al., 2019, 2020). WFMD-DOAS uses a least squares approach, which retrieves XCO from the shortwave infrared spectra recorded by the TROPOMI instrument. The TROPOMI WFMD CO retrievals (referred as WFMD hereafter) have undergone direct validation with independent reference data from the worldwide total carbon column observing network (TCCON, Wunch et al., 2011) which consists of ground-based Fourier transform spectrometer (FTS) instruments with a well-controlled light path. TCCON measurements are calibrated to the World Meteorological Organization (WMO) scale. As per this validation, WFMD XCO has a systematic error of 1.9 ppb and a random error of 5.1 ppb (Schneising et al., 2019).

### 2.1.2. Operational TROPOMI/SICOR CO product

The operational TROPOMI/SICOR CO product (referred to as SICOR hereafter) is retrieved using the Shortwave Infrared Carbon Monoxide Retrieval (SICOR) algorithm (Landgraf et al., 2016; Borsdorff et al., 2018a, 2018b). The validation study of SICOR with the CAMS data show a good agreement with global mean difference of +3.2% and a Pearson correlation coefficient of 0.97 (Borsdorff et al., 2018b) and for the Indian region, a 2.9% difference was found with CAMS with a standard deviation of 6% and a Pearson correlation coefficient of 0.9 (Borsdorff, 2018a). As per the validation of SICOR with ground-based total column measurements of TCCON, a mean bias of 6 ppb with a standard deviation of 3.9 ppb and 2.4 ppb has been found for clear and cloudy skies respectively (Borsdorff, 2018a).

### 2.2. Ground-level observations

To assess the model performance against the surface level measurements, we use measurements from ground-based air quality monitoring network maintained by the Central Pollution Control Board (CPCB) of India. The measurements of CO are performed using CO analysers based on non-dispersive infrared spectroscopy, and the data are provided as 6-hour averages via a publicly-accessible online portal (*https://app.cpcbccr.com/ccr/#/caaqm-dashboard-all/caaqm-landing/data*). Though we have analysed CO measurements available from all stations for the period of 3-20 November 2018, measurement stations that are too close to local emissions sources showing extremely large and ambiguous variations in which stability of the analyser may be questioned, were excluded for the evaluation. All the stations used for this evaluation are listed in Table 2.



## 3. WRF-Chem-GHG model

We utilize a high-resolution modelling framework based on a WRF-Chem-GHG (version 3.9.1.1, hereafter referred to as WRF) for simulating CO concentrations at a spatial resolution of 10 km × 10 km) and a temporal resolution of 1 hour. The model solves the compressible Euler non-hydrostatic equations and uses a terrain-following hydrostatic pressure coordinate system in the vertical direction (Skamarock et al., 2008). In our case, simulations have 39 vertical levels extending from the surface to 50 hPa (~20 km) and the model domain describes a region with a spatial extent of 3500 km × 2500 km, covering the Indian domain and some parts of Bangladesh, China, Nepal and Pakistan.

For meteorological initial and boundary conditions, we have taken ECMWF ERA5 data on an hourly basis with a horizontal resolution of 0.25°× 0.25°. For CO concentration fields, initial and boundary conditions are prescribed from the Copernicus Atmosphere Monitoring Service (CAMS re-analysis data). CAMS provides the estimated mixing ratios of CO with a spatial resolution of 0.25° × 0.25° at a temporal resolution of 6 hours on 60 vertical levels. For CO simulations, we have mainly used anthropogenic and biomass burning emissions tracers from external datasets. To represent anthropogenic contributions, we use the global EDGAR emission inventory (Emission Database for Global Atmospheric Research, version 4.3.2, the year 2012) data at a spatial resolution of 0.1°× 0.1°. EDGAR provides global inventories for GHG emissions and air pollutants on an annual basis, but we apply time factors in order to create hourly emissions. The time factors are based on the step-function time profiles published on the former EDGAR website: *http://themasites.pbl.nl/images/temporal-variation-TROTREP_POET_doc_v2_tcm61-47632.xls* (see Kretschmer et al., 2014; Steinbach et al., 2011, for further details). We use the CO emission data from the Global Fire Assimilation System (GFAS) for the year of 2018 to represent biomass burning emissions. GFAS is a satellite-based fire emission inventory (http://apps.ecmwf.int/datasets/data/cams-gfas/), which provides biomass-burning emissions daily at a global horizontal resolution of $0.1^o \times 0.1^o$. The inventory calculates the fire emissions by assimilating fire radiative power (FRP) observations from MODIS instruments on the polar-orbiting satellites Aqua and Terra, which observe the thermal radiation from fire activities at wavelengths around 3.9 µm and 11 µm (Kaiser et al., 2012). It achieves higher spatial and temporal (daily) resolution than almost any other inventory and can estimate near-real-time emissions. A number of studies have reported the underestimation of GFAS in fire emissions due to the limitations of the MODIS instruments, which do not capture all of the biomass burning emissions (Cusworth et al., 2018; Dekker et al., 2019; Huijnen et al., 2016; Kaiser et al., 2012; Mota and Wooster, 2018).

All these emissions fluxes are gridded to WRF's Lambert conformal conic projection grid with 10 km horizontal resolution, conserving the total mass of emissions. These fluxes are added to the first model layer and transported separately as tagged tracers (Pillai et al., 2016). In order to account for the CO transported from the boundaries, we used CAMS CO data derived at the boundary conditions and refer to this CO tracer as "background", meaning the concentration without considering any sources or sinks in the targeted domain. The total CO is then calculated as: CO total = CO background (BCK) + CO anthropogenic (ANT) + CO biomass (BBU).





Utilizing the emission tracers mentioned above as well as the multiple physics and chemistry
options and chemistry options and dynamics schemes, model simulations of CO are performed for the period 01–30
November 2018. The model setup does not include the deposition and chemical formation of
CO from volatile organic compounds (VOCs). Compared to the direct CO sources such as
anthropogenic and biomass burning emissions over the model domain, the indirect source from
VOC oxidation is much smaller, and the deposition processes are minor compared to the
transport of CO out of the model domain (Dekker et al., 2017). Also, the oxidation with the
hydroxyl (OH) radical is not considered. Based on a few sensitivity simulations, Dekker et al.,
(2017) reported a slight (4%) net decrease of enhancement when including chemical reactions
of CO and concluded that the CO enhancement pattern is hardly affected by VOCs and OH
oxidation.
**4. Methods**
**4.1. Comparison of WRF simulations with satellite column observations**
To evaluate the performance of WRF, we have performed a comparison study on a daily and
monthly basis using WFMD column CO (XCO) data during the period 1-30 November 2018
over the Indian domain. The WFMD dataset also provides the column averaging kernel vector
(AK), describing the vertical sensitivity of the retrieved CO column to the partial column at
different vertical levels (Schneising et al., 2019). In order to compare the satellite data with
model simulations quantitatively, we have to use the AK to take into account the vertical
sensitivity of the instrument. In the dataset, the elements of the AK mostly have values close to
1, meaning that the instrument is sensitive to the full column of CO. As such, the prior
estimates have a negligible contribution to the retrieved columns. To compare the simulated
concentration fields with the satellite observations, the simulated pressure-weighted column-
averaged dry air mole fraction after applying the averaging kernel, $c_{avgk}$ is calculated as
follows:
$$c_{avgk} = c + \frac{1}{m_0} + \sum_{l=1}^{n}(m_l(1 - A_l)\,(x_T{}^l - x^l))$$       Eq. 1
In this equation, $l$ is the index of the vertical layer and $n$ is the number of vertical layers, and $A_l$
the corresponding column-averaging kernel of the WFMD algorithm. $c$ is the pressure-
weighted column averaged dry air mole fraction calculated from model simulations. $x_T$ is the a
priori dry air mole fraction profile used by the WFMD retrieval algorithm, which is also
provided in the data product, and $x$ is the model simulation. $m_l$ is the mass of dry air for the
corresponding layer and $m_0$ is the total mass of dry air. For the comparison, we used only
WRF simulations that correspond to the satellite sampling time. For a fair comparison between
the satellite observations and model simulations, the averaging kernel matrix and a priori
profile for each retrieval have been applied to the corresponding model output as explained in
Eq. 1. For the ease of the statistical analysis, the observations, though comparable to the model
resolution, are gridded to the WRF spatial resolution of 10 km × 10 km. Both WFMD and
WRF averaged data for the month of November and a period of 6-9 November (enhanced
biomass burning period as per the GFAS data) are utilized in this study to investigate the





column enhancement by fire CO and their distribution over the study domain. During the enhanced biomass-burning period, a definite enhancement in XCO is found over the biomass burning hotspot. The monthly averaged map shows decreased concentration levels over these hotspots, which is attributed to the CO concentration dispersion resulted by changing weather conditions.

**4.2. Comparison of WRF simulations with ground-level observations**

To evaluate the model performance at surface level, we have performed a comparison study with the CO in situ measurements obtained from the ground-level pollution measurement stations. We use the data collected from 20 measurement stations within the IGP region and evaluation is done against each station data. In order to see overall agreement for different regions in the IGP, we have averaged the data temporally using only the stations within the corresponding regions (Delhi, Punjab, and the IGP). The entire month is not used here due to the existence of data gaps from several stations. In order to avoid very localised influence and noise in the observed data, the 1-hourly datasets are temporally averaged to 6-hourly resolution.

**5. Results and Discussions**

**5.1. Regional and seasonal variation of fire CO emission**

In order to examine the spatio-temporal variations of the monthly fire CO emission, we have divided the entire region into five sub-regions as shown in Fig. 1. The fire CO emissions show significant spatial and temporal variations, with predominant emissions over the Indo-Gangetic Plain (IGP), Central India (CI), and northeast India (NEI).

Figure 2 illustrates the integrated monthly fire CO emission for those regions in 2018. In most parts of India, the fire CO emissions peak during the March-April (pre-monsoon) period, accounting for about 76% of the annual emissions. This is consistent with a study based on the fire counts analysis from 1998-2009, which reported that more than 75% of the annual fires occurred during March-April (Sahu et al., 2015). Fire CO emissions during March are significantly higher when compared to other months, accounting for about 55% of the annual emissions for India. Although having a small geographical area, the fire activities over northeast India (NEI) made a significant contribution (57%) to emissions during pre-monsoon months, while the IGP contributed only about 5%. Central (CI) and southern regions (SI) of India add about 33% towards the pre-monsoon fire CO emissions, while North India (NI) shows fewer emissions during the whole year. However, emission spikes are seen in the IGP during the October-November (post-monsoon) period. Over the IGP, the fire CO emissions show evident monthly variations with a higher emission during the post-monsoon time compared to the pre-monsoon period. About 73% of the country's total fire CO emissions during the post-monsoon period are from the IGP region. Of these IGP post-monsoon emissions, 70% come from the northwest states of the IGP: Punjab and Haryana. Over this region, 25% of the total fire CO emissions happened within a short period during 6-9 November, which accounts for about 18% of the country's post-monsoon total fire CO emissions. During the monsoon time, all regions are found to have fewer fire emissions, which



can be attributed to the fact that rainfall leads to suppressed fire activity. In addition to the minimal possibility of fire activities during the rainy season, note that MODIS has only a limited capability to detect fire emissions over a cloudy scene.

The observed monthly variations in fire emissions are mainly due to factors such as post-harvest crop residue burning, meteorological conditions (dry weather), and land-use practices (Habib et al., 2006). The fire activities during post and pre-monsoon periods in India are mostly associated with the high-level crop residue burning during the post-harvest seasons (Sahu et al., 2015). Crop residue burning after harvesting is a general practice used by farmers to make the land clear for the next crop. Over the IGP, there are mainly two seasonal crop seasons known as Kharif (primarily rice), and Rabi (mainly wheat), which are harvested during post and pre-monsoon seasons respectively (Sahu et al., 2015). This results in the temporal variations of residue burning emissions over the IGP. Compared to other parts of the IGP, the northwest part of the IGP has the greatest preponderance of crop residues during the post-monsoon season (Singh and Panigrahy, 2011). Consistent with the spatial and seasonal differences in agricultural practices, we see a high level of fire CO emissions in this region during the short period of 6-9 November.

## 5.2. Enhanced XCO as observed by the satellite

Figure 3(a) shows the column CO dry mixing ratio retrieved from WFMD over the Indian domain averaged for the entire month of November and November 6-9 (most intense biomass burning period). During this period, higher values of column CO are observed over the northern part of India, particularly over the IGP region, compared to the other regions of India, showing higher values during the biomass burning period than the monthly average. A distinct enhancement in XCO can be observed during the biomass-burning period specifically over the state of Punjab and Haryana, with a distribution plume towards the southeast direction including the region of Delhi and Agra. Note that this emission hotspot is also seen in the GFAS inventory during the biomass-burning period (Fig 2). Consistency between the GFAS inventory and satellite observations suggest that the XCO enhancement over the northwest part of the IGP during 6-9 November can be attributed to the crop residue burning that occurred over the Punjab region. The consistency check between two retrieval products (WFMD and SICOR) has resulted in a very similar spatial CO pattern for both algorithms with a high correlation coefficient of 0.97 confirming the robustness of our findings between the two datasets over India (see Table 3). During early winter (November and December), the shallow PBL and low wind speed cause locally-emitted gases to be trapped in the lower atmosphere, which is considered to be the primary cause for high concentrations during this period. For a better understanding of the role of transport and CO emissions from biomass burning to the distribution over the domain, we utilized WRF model simulations and performed a comparison study with the WFMD observations as explained in Sect. 4.1.

## 5.3. Validation of WRF

### 5.3.1. Agreement with column observations



We compared WRF simulations with WFMD observations, averaged over the days of peak burning and over the full month of November 2018. Fig. 3 demonstrates these comparisons. Both satellite and the model show a higher level of column CO over the IGP region than over any other region of the domain. In the monthly averaged plots, the model slightly overestimates (by about 10 ppb) the XCO in most parts of the domain. Between the monthly averaged observations and the simulations, we find a mean difference of 7 ppb with a standard deviation of 8 ppb and a correlation coefficient of 0.87 (Fig. 4).

During the biomass-burning period, the model underestimates (by about 10-15 ppb) the enhancement over Punjab and some central parts of Uttar Pradesh while overestimating (by about 15-20 ppb) enhancements over the eastern parts of IGP including West Bengal and some parts of Bihar. Fig. 4 demonstrates these differences. Daily retrievals of WFMD and corresponding simulations for the biomass-burning period are shown in Fig. 5. An enhanced XCO is reported in both observations and simulations over the state of Punjab, starting from 6 November and gradually increases in the following days. During this period, the plume is seen to be partly transported in a southeast direction along the region of Delhi and Agra. Over the IGP, there exists an overall slight underestimation by WRF in comparison to TROPOMI during this period with a mean model-to-observation difference of -2.7 ppb.

Figure 6 shows the temporal evolution of the CO concentration in three cities (Barnala, New Delhi, and Agra) located along the transport pathway of pollution. The data are averaged in a 100 km x 100 km square around the centre of each city. During the biomass-burning period, the XCO over Barnala (Punjab) shows a steady positive increment with time with a peak on 9 November with a value of approximately 165 ppb. Both observations and simulations suggest a southeast transport of this plume that increases the CO concentration over Delhi and Agra during 8 and 9 November. Over Delhi, the WFMD XCO reached a maximum on 8 November while modelled CO showed a delay, with a maximum concentration on 9 November. On 9 November, observation shows more dispersed XCO over Delhi towards the southeast direction in comparison with model simulations. Over Agra, which is located far away from the pollution hotspot but along the transport pathway, an increase in XCO, which is consistent with that over the other two cities is found. The details in Table 3 confirm the minimal impact of differences in satellite retrieval algorithms on our results. This analysis suggests a promising usage of TROPOMI observations to understand the details of hotspot emissions and the distribution of transport. The model is able to capture many of these spatial and temporal patterns, supporting the potential use of WRF via inverse modelling to infer hotspot emissions using column measurements.

### 5.3.2. Agreement with ground-level observations

Figure 7 shows the model evaluation with ground-level measurements over the regions IGP, Delhi and Punjab for a period from 3 to 20 November 2018. The location of ground-level measurement stations used for this study is shown in Fig. 8. The entire month is not used here due to the existence of data gaps from several stations. Taking various ground-based stations over the IGP, Delhi and Punjab, we see an overall good agreement between model and measurements, with a correlation coefficient of 0.6 (for the IGP), 0.6 (Delhi) and 0.41



(Punjab). Among these three study regions, a lower correlation is found for the Punjab region in which measurement sites are very close to the biomass burning hotspots, therefore showing a larger variability compared to other stations. These variations are not fully reproduced by the model, resulting in lower correlations over Punjab region. Though the model is able to follow the temporal variation in the surface level CO concentrations, overall underestimations of 9 ppb and 54 ppb are found for Punjab and Delhi. For the IGP region, the model underestimates the observed enhancements considerably, resulting in a mean bias of 162 ppb. The observed underestimation of WRF can be attributed to the local source enhancements at the ground-level stations, which are closely located to the cities. For the Punjab region, the model CO surface concentration shows the influence of biomass burning starting from 6 November with a maximum of 800 ppb on 8 November. Unlike the Punjab region, the concentration patterns over Delhi and the IGP show a steadily increasing trend from 6 to 13 November, with a subsequent reduction in mixing ratios for the remaining days. Among these study regions during this period, the lowest and highest surface CO levels are observed over the regions Punjab (mean: 500 ppb) and Delhi (mean: 1500 ppb) respectively. Except for Punjab, we see better mean bias when excluding nighttime values (21 ppb for Delhi and 141 ppb for the IGP region), as the uncertainty from mixing height simulations is larger during nighttime compared to daytime. Surprisingly the overall underestimation increased in Punjab when using only daytime values, indicating a considerable underestimation of local emission sources, likely from the biomass emission inventory. Note that the GFAS fire emissions may be underestimated (Mota and Wooster, 2018). The GFAS fire emissions are partly based on the MODIS satellite instrument, and the limited resolution of the instrument misses many small fires, including biomass burning over India (Cusworth et al., 2018). Overall, the results show that the model simulation at a high spatial resolution is capable of capturing the CO enhancement and reduction pattern at most of the stations, however there is a non-trivial mean bias which can be attributed to issues with simulating transport and PBL dynamics in WRF as well as the variability in emission fluxes which is likely to be not sufficiently well represented in the emission inventories used.

**5.4. Contribution of different sources to the observed concentration**

To further investigate the contribution of different emission sources to the observations, we use the "tagged-tracer" option in WRF and separate the contributions from different sources as shown in Fig. 6 and 7. Note that the signals contributing to satellite observations are difficult to disentangle without underlying assumptions or the availability of multi-tracers such as CO and $NO_x$ and $NO_y$* ($NO_y$* includes $NO_x$, PAN, organic nitrates, $HNO_3$, and $N_2O_5$, e.g. Wang et al., 2002). The relative contributions of different emission sources and processes to the WRF CO column, as summarized in Table 4, clearly indicate the dominance of anthropogenic signals over biomass burning signals on the XCO enhancements. The significant impact of background signal owing to the advection from the domain boundary throughout the column indicates the influence of far-field fluxes and large-scale transport patterns on column CO (see Fig. 6). During the biomass-burning period, there exists a considerable contribution of biomass burning emissions to the column mixing ratios particularly over the Punjab region (14%). Relatively low contributions of biomass burning signals to the column in Delhi and the IGP compared to





Punjab indicates the dominant contribution of surface CO emission to the column in Punjab where the biomass emissions originated. It also suggests the possibility of less dilution of surface emissions during wintertime, enhancing the total column mixing ratios. The effect of advected biomass burning signals in terms of their contribution to the column can be seen over Delhi (12%), however this effect becomes smaller in the IGP (5%) due to further dispersion.

The diurnal variation in the surface level CO concentration pattern is due to the diurnal variation in the planetary boundary layer height (PBLH) combined with strong sources of CO at the surface. The contribution from emissions sources over the Delhi, the IGP and Punjab regions for the period of 3-20 and specifically 6-9 November are also summarised in Table 4. For all regions, the influence of background CO concentrations to the observed variability is minimal, as expected (see Fig. 7). The background influence is expected to be smaller for surface CO in urban areas where the CO fraction from local anthropogenic emissions dominates over the background signals. At ground level in Delhi and the IGP, a detectable enhancement in surface CO due to fire CO is found only during 6-9 November. During this period, the average contribution of biomass burning to the ground level concentration is 10%, while the anthropogenic contribution is 79-83%. During 3-20 November over Delhi, however, the average contribution from fire dropped to 4% compared to 85% in the case of the anthropogenic contribution.

Overall, our findings suggest that the enhanced CO levels during pollution episodes over Delhi and the greater part of IGP are affected by biomass burning. However, a more significant contribution comes from anthropogenic emissions. Unlike the surface CO mixing ratios, the majority of the column CO mixing ratio is contributed by the background signal. A recent study conducted by Dekker et al., (2019) concluded that there exists an underestimation in GFAS fire emission data over the Indian region, which is supported by our findings. However, over the Punjab region, biomass burning played a significant role in determining the ground level CO measurements, especially during 6-9 of November, during which enhanced fire activities occurred. This has contributed considerably to the column mixing ratio that is detected by TROPOMI. On average, for 3-20 of November, 17 % of the total ground level CO concentration over the Punjab region are on account of fire CO emission, whereas for 6-9 November, the share is about 38%.

**5.5. Effect of meteorology**

Usually pollution episodes during winter are the result of meteorological conditions due to low wind speed and shallow boundary layer (PBL height). Figure 9 demonstrates the influence of the PBL height and surface-level wind speed to the observed CO level. We found a negative correlation of CO with modelled PBLH (-0.83 (IGP), -0.73 (Delhi), -0.56 (Punjab)) and wind speed (-0.62 (IGP), -0.40 (Delhi), -0.24 (Punjab)). A strong negative relation between PBL height and CO level is seen, indicating the impact of meteorology on the diurnal variation of surface-level CO concentration. Among the regions, a less negative correlation of CO with PBLH and wind speed is observed for Punjab. It suggests that when compared to Delhi and the IGP, the surface level CO variation over the Punjab region cannot be explained by meteorology



alone: Here the local emission activities, such as biomass burning, explain more of the variability in surface level CO.

A gradual increase in surface CO levels was observed from 3 to 13 of November during which an overall decrease in PBL height and surface-level wind speed took place. The highest CO values around Delhi were found during 11-13 November, just before the winds and PBL height were increasing. The findings suggest that the meteorological conditions have a large impact on the surface level CO concentration, especially over the IGP and Delhi. Our results are consistent with Dekker et al. (2019), who identified that the meteorological conditions contributed significantly to the enhancement of CO mixing ratios at the ground level during November 2017. Similarly Kariyathan et al. (2020), by using a-temporal emission fields and a Lagrangian modelling framework, found a considerable impact of meteorological conditions during November 2017 that contributed to the enhancements of trace gases over Delhi. Together with strong emissions (anthropogenic and biomass burning), they found that these enhancements could be several orders of magnitude higher compared to other seasons.

## 6. Conclusions

The Tropospheric Monitoring Instrument (TROPOMI) on board ESA's Copernicus Sentinel-5 Precursor (S5P) mission provides shortwave infrared measurements of CO with daily global coverage and a high spatial resolution of $7\times7$ km$^2$. These high density and high accuracy CO column observations enable us to investigate high CO pollution episodes over India, which otherwise would not have been possible at this spatial resolution. In this study, we demonstrate the usefulness of TROPOMI CO column observations for detecting and analysing local CO enhancement over India during winter 2018, employing WRF at a resolution comparable to TROPOMI to aid in the interpretation of the data. The GFAS biomass burning emission product shows a substantial amount of fire CO emitted from various parts of India during the year 2018. Over the IGP, the fire CO emission shows an apparent monthly variation with a higher emission during the post-monsoon time compared to the pre-monsoon period. A large amount of fire CO emissions is reported over the state of Punjab within the short period of November 6-9 2018. Consistent with the emission data, TROPOMI XCO shows a clear enhancement during November not only over the fire emission hotspots but also along the western parts of the IGP, including the national capital of India, Delhi.

For further analysing the causes of these enhancements, we used simulations generated by WRF. A similar study conducted by Dekker et al., (2019) also utilized WRF for identifying the sources contributing to the high pollution event in North India during 2017, but using preliminary TROPOMI data generated with the SICOR algorithm. The present study uses both fossil fuel emissions data based on EDGAR (version 4.3.2) with hourly variations and biomass burning emissions data based on GFAS fire CO emissions. If WRF reproduces the transport sufficiently well, the mismatch between the simulations and observations is mostly caused by uncertainties in the prior emission fluxes (EDGAR and GFAS) due to the linear dependence of the CO concentrations on the source strength of the emissions. To evaluate the simulated CO fields with the observed CO columns, we applied the WFMD averaging kernel to the corresponding model profile, taking into account the





vertical sensitivity of the satellite measurement. Overall, we find a good agreement between WRF and WFMD with a mean difference of 7 ppb, a standard deviation of 8 ppb, and a spatial correlation coefficient of 0.87.

Our analysis shows that daily observations from TROPOMI allow pollution transport from the emissions hotspots to be captured. As an example, we analysed the pollution transport from the fire emissions hotspots over northern India during the enhanced burning period of November 6-9. WFMD XCO level started to rise over the fire emission hotspots from 6 November and gradually increased during the following days. Both WFMD and model simulations show the transport of CO polluted air masses towards the northeast part of the IGP along with the capital city Delhi. Due to this pollution transport, the CO concentration level in the cities along the transport pathway shows CO enhancements. A similar transport pattern is also observed in our WRF model simulation. This supports the reliability of WRF transport simulation and suggests the potential of using WRF to estimate CO emission via flux inversions. The good agreement between WFMD and SICOR retrievals over India confirms the robustness of our findings irrespective of the differences in the retrieval algorithm.

For the further evaluation of WRF with surface measurements, we used ground level CO measurements from the stations along the IGP for the period of 3-20 November 2018. The comparison shows a good agreement between simulations and observations with a correlation coefficient of 0.6 (for the IGP), 0.6 (Delhi) and 0.41 (Punjab). Over these regions, the surface CO showed a steady increasing trend from 6 to 13 November, followed by a reduction in mixing ratio in the following days. Among these study regions, the lowest and highest surface CO level was observed over the regions Punjab and Delhi respectively.

Overall, our results imply a minimal role of biomass burning in terms of its contribution to both column and surface enhancements compared to other anthropogenic sources, except for the state of Punjab during the high pollution episodes. This is also consistent with Dekker et al. (2019), which concluded that the low wind speeds and shallow atmospheric boundary layers were the most likely causes for the temporal accumulation and subsequent dispersion of CO during the biomass-burning period in Novemebr 2017. By comparing our results with Dekker et al. (2019), we can infer the significant role of atmospheric dynamics and anthropogenic emissions on producing exorbitant level of pollutants and trace gases during every winter in northern India. While these anthropogenic urban sources (e.g. road traffic, residential air-conditioning systems, industries and power plants) are primarily responsible for the CO enhancement in winter months, there exists a non-trivial fraction of contribution from biomass-burning activities in Punjab and nearby locations for a short duration of time. The variation in surface-level CO concentrations is found to be influenced significantly by the meteorological parameters such as PBL height and surface level wind speed. Our results show a clear influence of atmospheric transport leading to a complex CO enhancement pattern. This demonstrates the need for high-resolution models in the interpretation of TROPOMI observations in order to get more insight into the pollution transport and deduce causes for the observed enhancements (and resulting poor air quality) over India.



In an effort towards minimizing the pollution episodes, a robust evaluation of emissions inventories and their trends is vital, particularly in light of uncertainties in existing emission sources, and the limited availability of appropriate emissions estimates in different emission sectors. Studies identifying the emissions hotspots and understanding their transport patterns such as that carried out in Dekker et al. (2019) and in this work are thus important for further decision making for emission control. While WRF is able to reproduce observations reasonably well, the model errors are not negligible when utilizing TROPOMI observations for emission estimates. Nevertheless, we emphasize the importance of taking rigorous policy measures to reduce residential and commercial emissions in addition to measures already being taken in the agricultural sectors (e.g. the implementation of second-generation direct-seeders, such as the Happy Seeder, which facilitate sowing under heavy stubble conditions, thereby avoiding the need for residue burning, NAAS, 2017). The future task involves the implementation of appropriate inverse techniques suitable for flux inversion of spatially resolved sources of CO emissions over India.



## 1 Code/Data availability

The WRF-CO model simulations used in this study are available upon request to the corresponding author D. Pillai (*dhanya@iiserb.ac.in*, *kdhanya@bgc-jena.mpg.de*). The WRF-Chem source code is publicly available (https://ruc.noaa.gov/wrf/wrf-chem/). The input data used for simulations in this study are either publicly available or available upon request to D. Pillai. The S5P WFM-DOAS data can be accessed from *http://www.iup.uni-bremen.de/carbon_ghg/products/tropomi_wfmd/* and the operational product is available at *https://scihub.copernicus.eu/*. The ground-based CO data analysed in this study can be accessed from *https://app.cpcbccr.com/ccr/#/caaqm-dashboard-all/caaqm-landing/data*.

## 10 Author Contribution

DP designed the study and performed the model simulations. AV and DP interpreted the results. AV performed the TROPOMI/WFMD and in-situ data analysis, and wrote the paper. JM, CG, MB, and OS provided significant input to the interpretation, and the improvement of the paper. All authors discussed the results and commented on the paper.

## 15 Competing interests

The authors declare they have no conflict of interest.

## 17 Acknowledgements

This study is supported by the funding from the Max Planck Society allocated to the Max Planck Partner Group at IISERB. We acknowledge the support of IISERB's high performance cluster system for computations, data analysis and visualisation. The WRF-Chem simulations were done on the high performance cluster Mistral of the Deutsches Klimarechenzentrum GmbH (DKRZ). The first author acknowledges the research infrastructure support provided by IISERB and thanks Thara Anna Mathew and Monish Deshpande for their contribution to graphics.





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



1    Table 1. Overview of the TROPOMI CO products used in this study

| Data ID | Satellite Data | Retrieval algorithm | Data access | Reference |
|---|---|---|---|---|
| WFMD | TROPOMI/WFMD CO | Weighting Function Modified Differential Optical Absorption Spectroscopy (WFM-DOAS) | (*http://www.iup.uni-bremen.de/carbon_ghg/products/tropomi_wfmd/*) | (Schneising et al., 2019, 2020) |
| SICOR | TROPOMI/SICOR CO | Shortwave Infrared Carbon Monoxide Retrieval (SICOR) | (*https://scihub.copernicus.eu/*) | (Landgraf et al., 2016; Borsdorff et al., 2018a, 2018b) |





1    Table 2. List of ground-level measurement stations used for this study

| No | Station name | State | Latitude (°N) | Longitude (°E) |
|----|--------------|-------|---------------|----------------|
| 1 | Hardev Nagar, Bathinda - PPCB | Punjab | 30.23 | 74.90 |
| 2 | Civil Line, Jalandhar - PPCB | Punjab | 31.32 | 75.57 |
| 3 | Ratanpura, Rupnagar - Ambuja Cements | Punjab | 30.00 | 76.60 |
| 4 | NISE Gwal Pahari, Gurugram - IMD | Punjab | 28.42 | 77.14 |
| 5 | Burari Crossing, Delhi - IMD | Delhi | 28.72 | 77.20 |
| 6 | Delhi | Delhi | 28.55 | 77.25 |
| 7 | IGI Airport (T3), Delhi - IMD | Delhi | 28.56 | 77.11 |
| 8 | ITO, Delhi - CPCB | Delhi | 28.62 | 77.24 |
| 9 | Lodhi Road, Delhi - IMD | Delhi | 28.59 | 77.22 |
| 10 | NSIT Dwarka, Delhi - CPCB | Delhi | 28.60 | 77.03 |





| 11 | Patparganj, Delhi - DPCC | Delhi | 28.62 | 77.28 |
|---|---|---|---|---|
| 12 | Sector - 125, Noida - UPPCB | Utter Pradesh | 28.50 | 77.30 |
| 13 | Sanjay Palace, Agra - UPPCB | Utter Pradesh | 27.20 | 78.00 |
| 14 | Central School, Lucknow - CPCB | Utter Pradesh | 26.88 | 80.93 |
| 15 | Ardhali Bazar, Varanasi - UPPCB | Utter Pradesh | 25.40 | 82.90 |
| 16 | IGSC Planetarium Complex, Patna - BSPCB | Bihar | 25.60 | 85.10 |
| 17 | Ghusuri, Howrah - WBPCB | West Bengal | 22.61 | 88.34 |
| 18 | Padmapukur, Howrah - WBPCB | West Bengal | 22.56 | 88.27 |
| 19 | Rabindra Bharati University, Kolkata - WBPCB | West Bengal | 22.62 | 88.38 |
| 20 | Victoria, Kolkata - WBPCB | West Bengal | 22.54 | 88.34 |



Table 3. Comparison between WFMD and SICOR products over India during the burning
period and the full month of November 2018. Abbreviations N, MB, SD, and R correspond to
the number of observations, mean bias, standard deviation of differences, and correlation
coefficient respectively.

| Peak Burning Period Only (6-9 November 2018) | N (SICOR): 93416<br>N (WFMD): 98093<br>MB (SICOR-WFMD): 1.85 ppb<br>SD (SICOR-WFMD): 4.86 ppb<br>R (SICOR vs. WFMD): 0.97 |
|---|---|
| All of November 2018 | N (SICOR): 555724<br>N (WFMD): 638215<br>MB (SICOR-WFMD): 1.72 ppb<br>SD (SICOR-WFMD): 4.27 ppb<br>R (SICOR vs. WFMD): 0.97 |





1   Table 4. Contribution from different emissions sources to the CO concentration at ground level
2   (GL) between 6-9 and 3-20 November 2018. Abbreviations ANT, BBU and BCK represent
3   anthropogenic, biomass burning, and background signals respectively (see Sect. 3)

| Period | CO | Delhi | | | Punjab | | | IGP | | |
|---|---|---|---|---|---|---|---|---|---|---|
| | | ANT | BBU | BCK | ANT | BBU | BCK | ANT | BBU | BCK |
| 6 – 9 November 2018 | Column | 35 % | 12 % | 53 % | 21 % | 14 % | 65 % | 32 % | 5 % | 63 % |
| | Surface | 83 % | 10 % | 7 % | 49 % | 38 % | 13 % | 79 % | 10 % | 11 % |
| 3 – 20 November 2018 | Column | 43 % | 6 % | 51 % | 25 % | 8 % | 67 % | 34 % | 3 % | 63 % |
| | Surface | 86 % | 4 % | 10 % | 60 % | 17% | 23 % | 82 % | 4 % | 14 % |


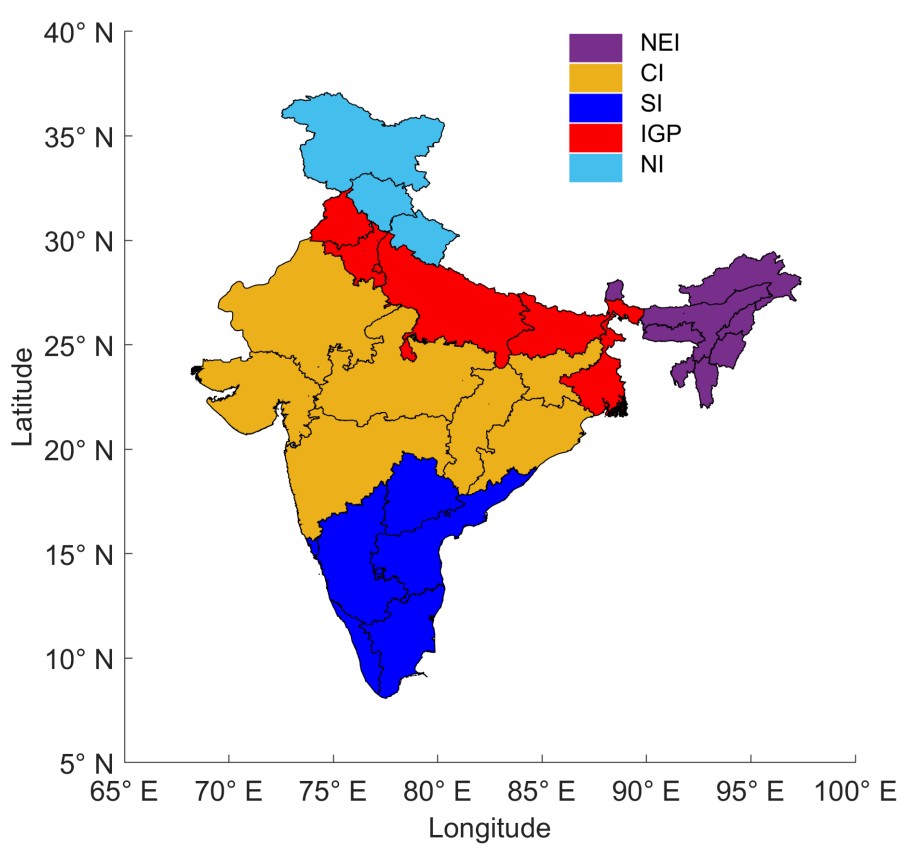

**Figure 1** India partitioned into five different areas for analysis: northeast India (NEI), central
India (CI), southern India (SI), the Indo-Gangetic Plain (IGP), and northern India (NI).





(a)                                                                    (b)

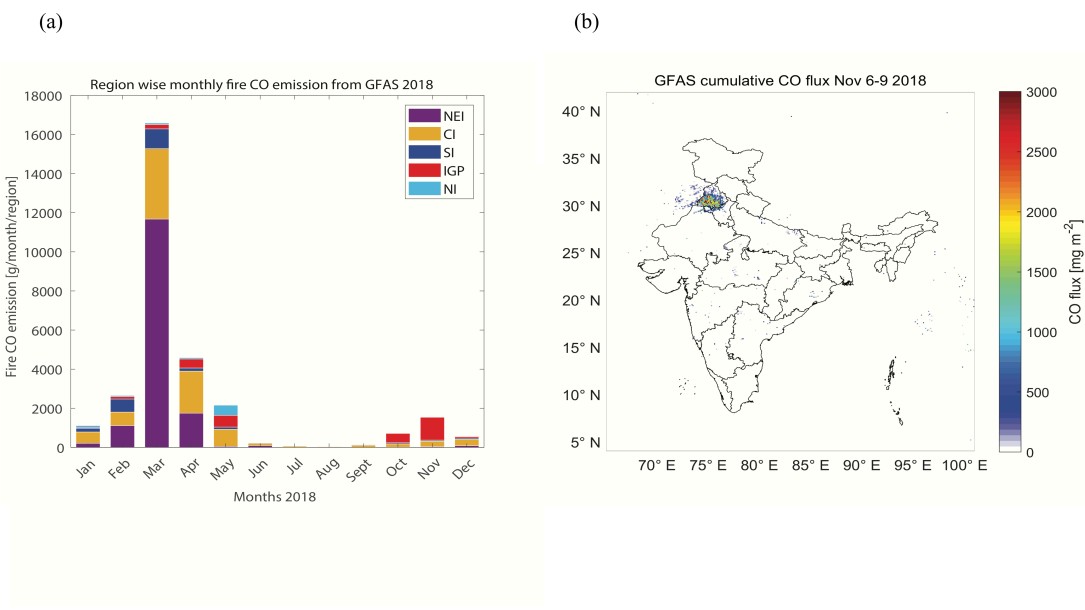

**Figure 2 (a)** The monthly integrated GFAS fire CO emissions (mg/m$^2$/month) over different
regions of India (as seen in Figure 1) during the year 2018. **(b)** Integrated GFAS fire CO
emission during 6-9 November 2018.

(a)TROPOMI/WFMD

(b)WRF

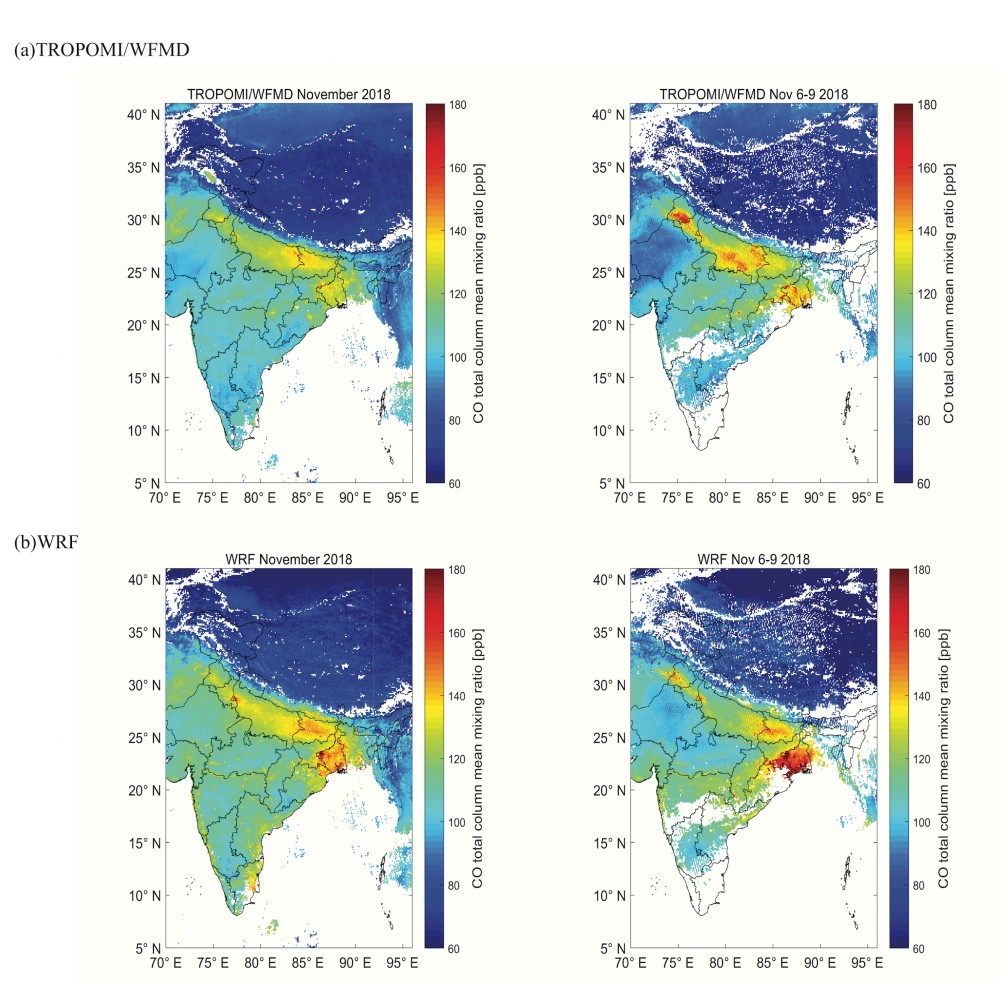

2
**Figure 3:** CO total column mixing ratios averaged over all of November 2018 (left panel) and
from 6-9 November 2018 (right panel). **(a)** TROPOMI/WFMD **(b)** WRF model



(a)                                          (b)

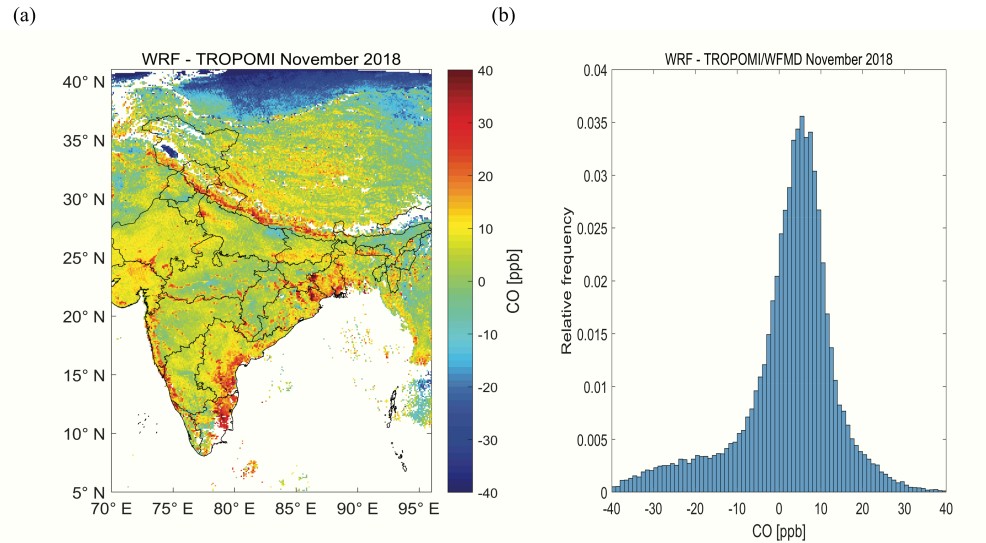

2      **Figure 4: (a)** Differences of CO total column mixing ratios (WRF –TROPOMI/WFMD)
3      averaged over the month of November 2018. **(b)** Histogram of the differences (Mean bias: 7
4      ppb; standard deviation: 8 ppb; correlation coefficient: 0.87).



(a)  (b)

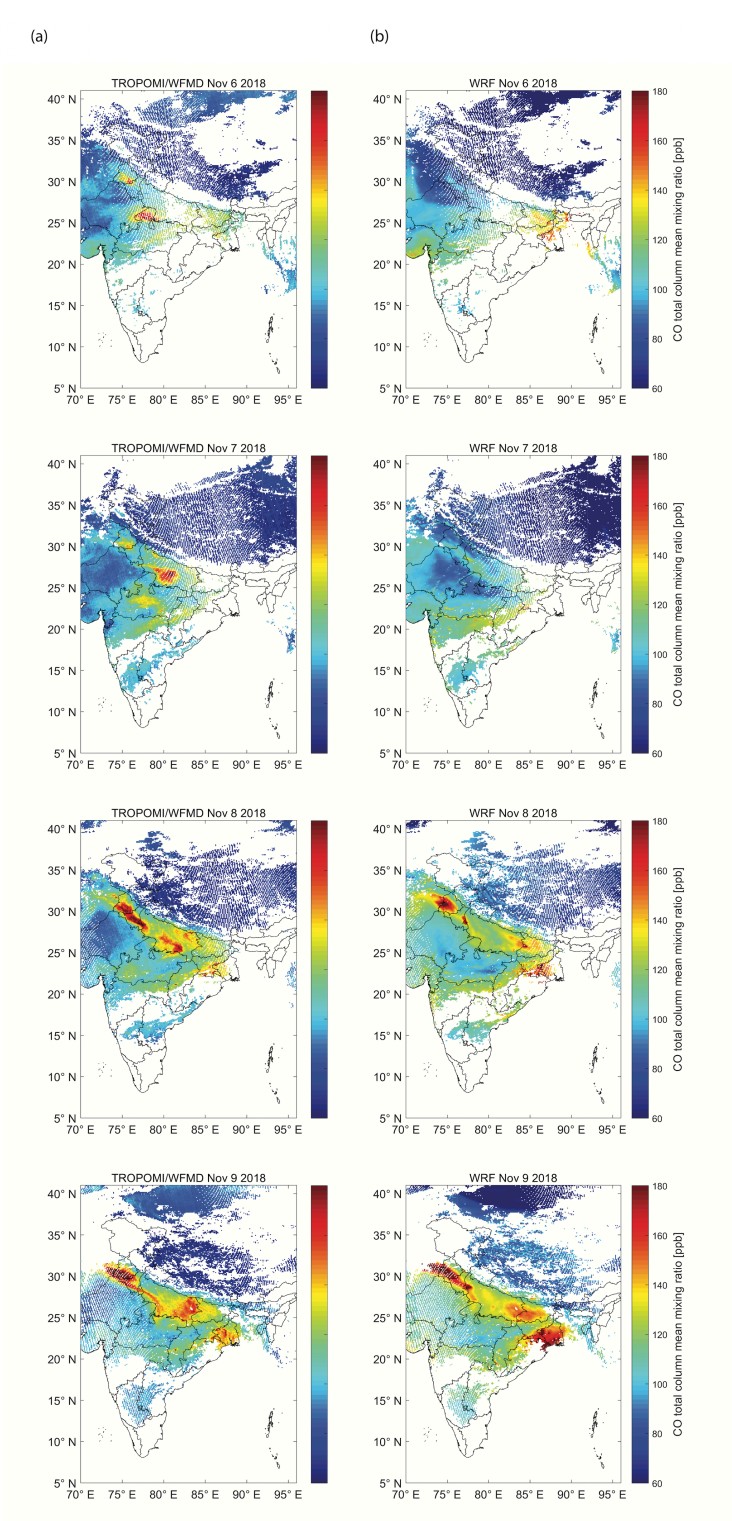



1 **Figure 5: (a)** Daily column CO observations from TROPOMI/WFMD (left panels) and **(b)**
2 collocated WRF simulation (right panels) for November 6-9 2018.





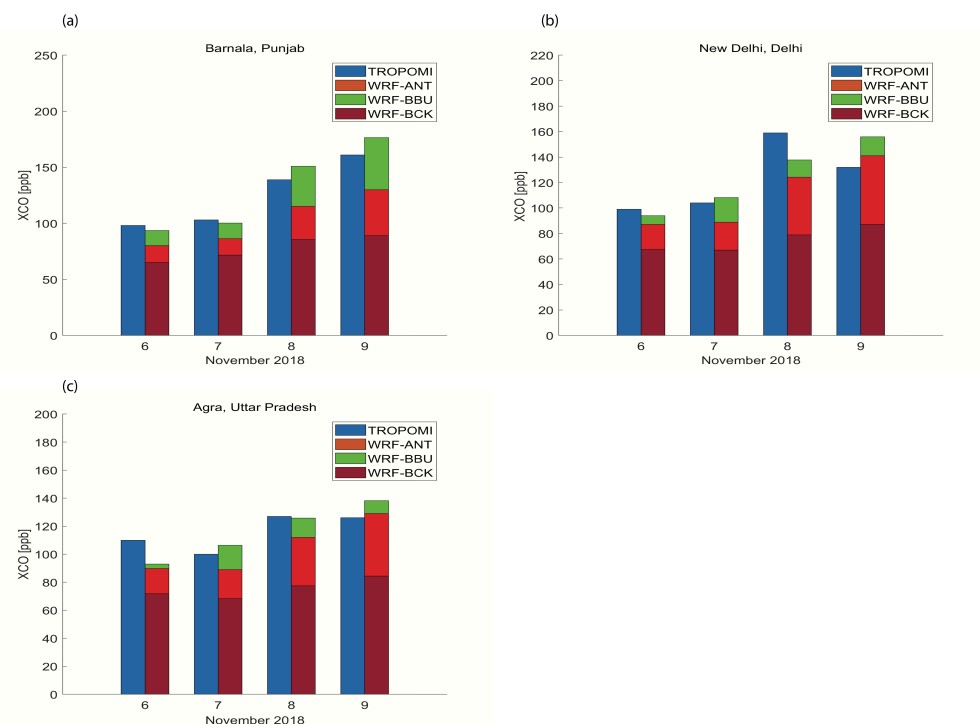

2 **Figure 6:** Carbon monoxide (CO) total column mixing ratios over **(a)** Barnala, **(b)** New Delhi,

3 and **(c)** Agra for individual days from 6–9 November 2018.





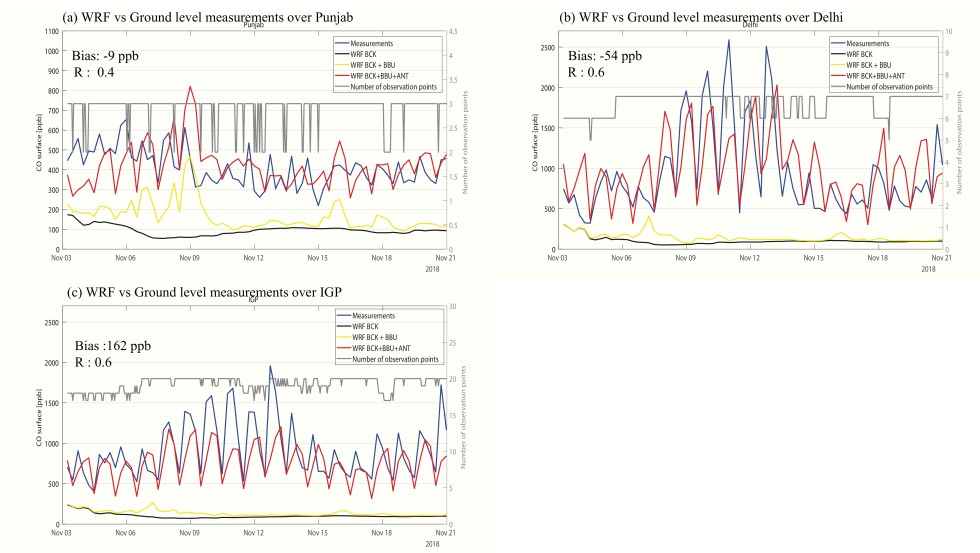

2    **Figure 7:** Ground level CO measurements and WRF model simulations for a period of 3-20
3    November 2018 over **(a)** the IGP region **(b)** Delhi **(c)** Punjab.





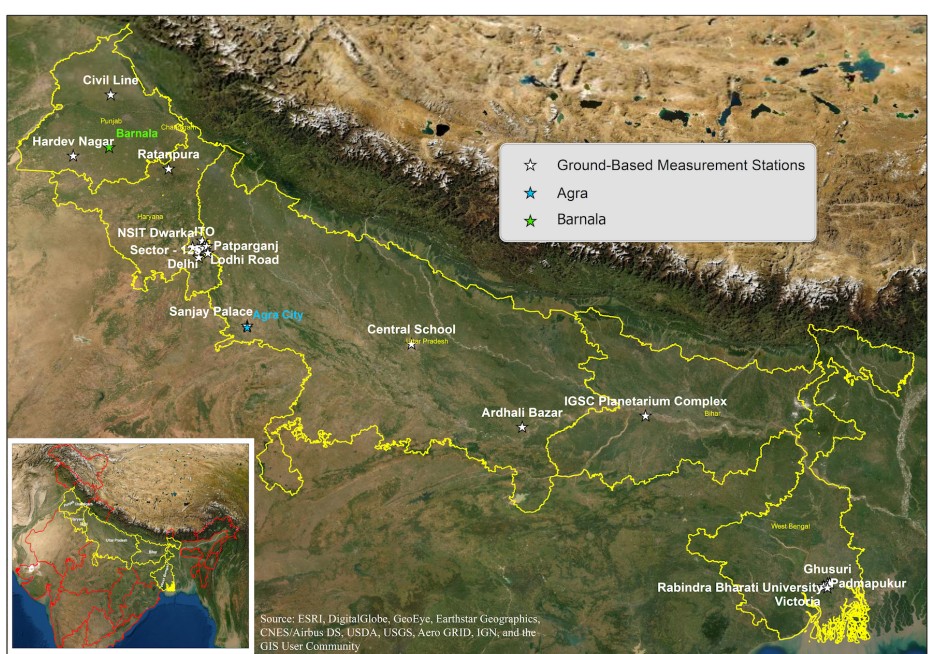

2 **Figure 8:** Map showing the locations of sites used for model evaluation. The yellow contour
3 represents the IGP region. The inset image shows the broader region for context.





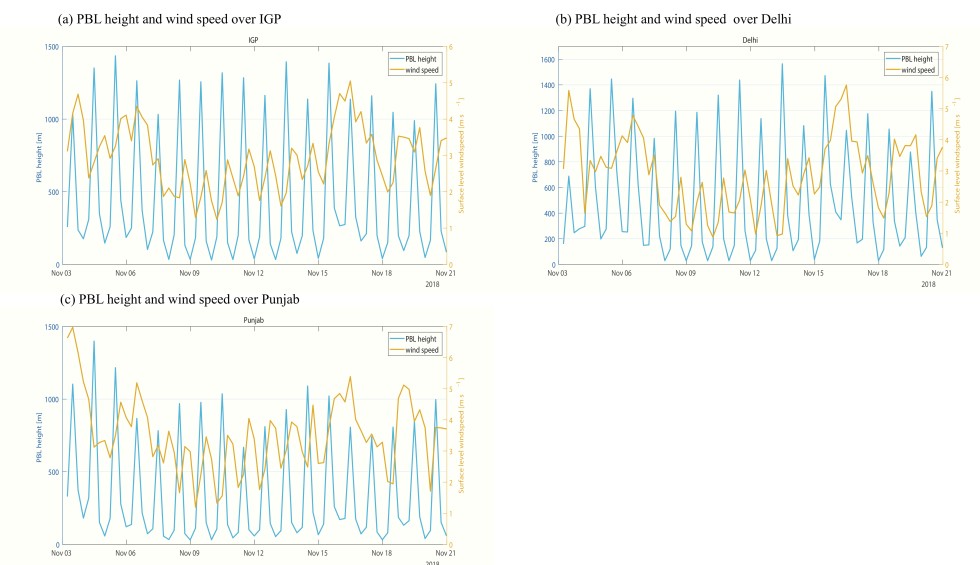

**Figure 9:** PBL height and surface level wind speed from WRF model simulations for a period
of 3-20 November 2018 over **(a)** the IGP region, **(b)** Delhi, and **(c)** Punjab.

