# Peer review of "Using satellite measurements and mesoscale modelling to understand the contribution to an extreme air pollution event in India"

_Atmospheric Chemistry and Physics, 2020_

## Referee Comment (RC1) · Anonymous Referee #1 · 3 Dec 2020

Review

Atmospheric Chemistry and Physics

Title

Using satellite measurements and mesoscale modelling to understand the contribution to an extreme air pollution event in India

Authors

[Figure]

Ashique Vellalassery, Dhanyalekshmi Pillai, Julia Marshall, Christoph Gerbig, Michael Buchwitz, and Oliver Schneising

Summary

This paper analyses the contribution of biomass burning emissions, anthropogenic emissions, and meteorology to carbon monoxide (CO) concentrations across the Indo−Gangetic Plain (IGP) in India. A case study focuses on an air pollution episode during November 2018, coincident with large agricultural biomass burning across the states of Punjab and Haryana. This study quantifies these contributions using chemical transport model simulations with tracers and evaluates the model using uses high spatial−resolution satellite measurements. Air pollution exposure is an important public health problem in India, and acute episodes can be particularly severe in winter. The topic of this paper is relevant to the scope of Atmospheric Chemistry and Physics.

My main criticisms are to increase the model description, discuss specific anthropogenic sources, and consider particulate air quality.

A substantial portion of the paper is dedicated to evaluating the model skill in simulating CO concentrations. However, the paper does not mention which physics, chemistry, aerosol, and dynamics schemes were used. A discussion of the gas−phase chemistry mechanism would be especially relevant. The authors find a larger contribution from anthropogenic emissions than from biomass burning emissions to CO concentrations across the IGP, except within the Punjab for a short period of time during the episode. Examples of these anthropogenic emissions are given (e.g. residential air conditioning systems). However, alternatives to these may be expected to be more likely (e.g. residential solid fuel use for cooking and heating, coal−fired brick kilns). Hence, the chosen examples need explaining.

Air quality in India is mainly important in terms of fine particulate matter (PM2.5) exposure and these biomass burning events also contribute significantly to ambient PM2.5 concentrations (India State-Level Disease Burden Initiative Air Pollution Collaborators

2019, Cusworth et al 2018, Jethva et al 2019). It would be useful to see a discussion of how these episodic emissions contributed to PM2.5 exposures and the associated acute health impacts.

Overall, this paper provides an interesting analysis of how biomass burning contributes to CO concentrations during a winter air pollution episode in India. The paper would be improved by adding model details and further discussing its implications.

Comments

- 1. Title: The contribution of what? Also, is mesoscale modelling the most accurate term here?

- 2. This paper has many acronyms. Are all these necessary?

- 3. The paper aims to address 5 questions. It would be useful to have a concise summary of the answers to these questions in the conclusion.

- 4. Page 2 line 10, page 3 lines 3 and 10, page 4 lines 21 and 40, page 5 lines 13, 14, and 30, page 6 lines 1, 9, 10, and 23, page 7 lines 10, 19, and 28, and page 12 line 34: Define acronyms at first use.

- 5. Lines 8−13: Why is the fire−radiative power (FRP) approach more accurate than inventory approaches? In the second half of this sentence, the authors mention inaccuracies in these derived approaches. If by derived the authors imply FRP, then this would be useful to clarify for the reader.

- 6. Page 4 line 36, page 5 line 22, page 7 line 27, and page 14 lines 17 and 18: Acronyms already defined.

- 7. Page 5 lines 9−19: This is a very long sentence, which requires shortening. Also, this comparison to previous work may be better suited to the Discussion.

- 8. Page 5 line 40, and page 6 line 12: Typo: WFM−DOAS.

- 9. Page 13 line 36: Typo: PBL.

- 10. Page 15 line 31: Typo: November.

- 11. Page 16 line 10: The authors emphasise residential and commercial emissions. This needs to be more specific i.e. are they implying residential solid fuel use emissions from cooking and heating?

- 12. Table 2, Figure 2, Figure 3, Figure 4, and Figure 5: Define acronyms in captions.

- 13. Figure 2a: Unit mismatch between caption (mg/m2/month) and $y-$axis label (g/month/region).

References

Cusworth D H, Mickley L J, Sulprizio M P, Liu T, Marlier M E, DeFries R S, Guttikunda S K and Gupta P 2018 Quantifying the influence of agricultural fires in northwest India on urban air pollution in Delhi, India Environ. Res. Lett. 13

India State-Level Disease Burden Initiative Air Pollution Collaborators 2019 The impact of air pollution on deaths, disease burden, and life expectancy across the states of India: the Global Burden of Disease Study 2017 Lancet Planet. Heal. 3 e26–39

Jethva H, Torres O, Field R D, Lyapustin A, Gautam R and Kayetha V 2019 Connecting Crop Productivity, Residue Fires, and Air Quality over Northern India Sci. Rep. 9 16594 Online: http://www.nature.com/articles/s41598-019-52799-x

---

## Referee Comment (RC2) · Anonymous Referee #2 · 4 Dec 2020

General Description and Comments:

This paper describes the application of combining satellite observation of CO with WRF-Chem modeling to better understand the transport patterns and contribution of various sources to high CO concentration in North India in November 2018. WRF-Chem model outputs are compared against XCO from TROPOMI and surface CO concentration from ground-based stations. CO from biomass burning source, anthropogenic source, and background are tagged separately in the WRF-Chem simulation to assess the contribution of each source to total CO. The authors state that there is

a good consistency between TROPOMI XCO observation and modeled values. Lower agreement was observed between ground stations and model values due to higher sensitivity of surface concentration to meteorology variables such as PBL and wind speed. Authors state that WRF-Chem captured the transport pattern well during this period but fail to provide enough evidence for this. They found minimal role of CO from biomass burning sources to the total column and surface CO enhancements except in regions close to biomass burning sources. Finally, they emphasize the importance of mitigation policies that focus mostly on controlling anthropogenic sources because of the significant impact of these sources on regional air quality.

In my opinion, the paper is well-written and well-organized. However, I did not find enough evidence for the major conclusions in the paper. I would recommend this paper for publication only after major revision and adding more discussion on the transport of plume in the model and the model performance in capturing these patterns. The uncertainties (either in emission or transport) need to be discussed further. Some ideas on how to strengthen your argument:

- Adding backtrajectory analysis can help better understand the transport pattern in the model. - A perturbation run with increased fire emission can help examine your conclusions further and also gives an insight into the transport error of the model. Or using other fire emission inventories - Cross section plots can help with understanding the transport patterns.

General comments: - Add more details of WRF-Chem model configuration. Any nudging or re-initialization of the model? By running the model freely for a month will increase the errors in the model.

- There is no model performance evaluation with respect to meteorology variables. Please consider adding it to the discussion.

- Even after reading P5 L11-19, I still think that this study is very similar to Dekker et.al 2019. Consider adding more analysis to your study.

- Consider adding evaluation of fire emission performance. Perhaps compare the emission with VIIRS active fire points to see how emission inventory underestimated CO emission. What about other fire emission inventories?

- Have you looked at the performance EDGAR inventory in this region in the literature? Can errors in EDGAR partly justify the large errors in surface CO?

- Map of the contribution of emission CO source to total in November 2018 can be helpful.

Specific Comments

P5 L15 – I did not find (2) in the paper. Where did you examine the regional distribution of CO for the entire year? You also did not specify which model configuration you used and why you used these options.

P6 L40 – I recommend using a map marking station locations rather than listing stations in Table 2. Perhaps replacing Table 2 with Figure 8.

P7 L2-9 – Add more details about the WRF-Chem configuration and options.

P7 L34 – consider adding Pan et al 2020 reference.

P7 L37 – So all emissions from fires are released from the surface? How does this limit the model performance?

P8 L1 – Multiple physics and chemistry options and dynamics schemes are not discussed in the paper.

P9 L22-26 – consider referencing Kulkarni et al. 2020 here on fire activities

P10, sec 5.3.1. – It is difficult to see the differences between model and obs in Figure 3. Please add bias map for Nov 6-9 (similar to Fig 4 a). Perhaps you may want to reorganize the panels in fig 3 and 4. Consider using different color bars for bias maps and XCO maps.

P11 L4-7 – You have a relatively large domain and you are focusing in a smaller region in the domain with high biomass burning activities. Averaging the biases over the whole domain cannot really help with your discussion. I suggest providing statistics only for IGP region.

P11 L8 – by "biomass-burning period" do you mean the month of November or just Nov 6-9? Fig 4 only shows November.

Figure 5 – add another column with CO emission from fire similar to Fig 2-b but for each day. Discuss how fire emission varied day by day and if model captured it.

P11 L17 – Looking at Fig 5 it looks like on Nov 6 and 7 model didn't capture emissions correctly and on Nov 8 model didn't transport the pollutant correctly. Please discuss the errors further. There are other papers that looked at a similar problem for Nov 2018 such as Kumar et al., 2020 and Roozitalab et al., 2020, please compare your findings with their conclusions.

P11 L 18 – Fig 6 Can you mark these locations on the map on Fig 5? Please use same XCO range for all three sub plots.

P11 L18-34 – Adding backtrajcetory analysis can be beneficial for the argument here. More discussion on model errors in capturing transport pattern is needed. Before doing inverse modeling to constrain emission you need to understand the model errors.

P11- L35 – Are Punjab and Delhi region showed in Fig 6 and 7 the same regions? I suggest marking these regions on the map for both column and surface observations.

P12 L3 – What do you mean by larger variability? To me, it looks like obs CO variability is lower in Punjab compared to Delhi

P12 L28 – From Fig 5 it looks like model did not have any fires near Punjab on Nov 6 and Nov 7. You can add a similar discussion in this section by referring to fig 5. Also adding backtrajectory analysis can help with understanding the transport patterns.

P13 L10 – What do you mean by observed variability? The influence of background is not minimal when looking at column CO.

P13 L20 – Since GFAS underestimates fire emissions do you expect a higher contribution of fire CO to total CO in Punjab in reality?

P13 L32 – Please overlay obs surface CO and model in these plots. Also, use the same range for Y axis for all plots. Comparing with observed meteorology variables can greatly benefit the discussion here and help justify the large biases of surface CO for some days.

P14 L1 – You got to this conclusion based on correlation numbers from Nov 3-20 and I don't think that is enough. How are the correlations during Nov 6-9?

P14 L15 – The level of contribution of meteorology to regional air quality can vary day by day. Have you looked at other studies that looked at Nov 2018 such as Kumar et al., 2020 and Roozitalab et al., 2020?

Reference: Kumar, R., Ghude, S. D., Biswas, M., Jena, C., Alessandrini, S., Debnath, S., Kulkarni, S., Sperati, S., Soni, V. K., and Nanjundiah, R. S.: Enhancing Accuracy of Air Quality and Temperature Forecasts During Paddy Crop Residue Burning Season in Delhi Via 775 Chemical Data Assimilation, Journal of Geophysical Research: Atmospheres, 125, e2020JD033019, 2020.

Roozitalab, Behrooz, Gregory R. Carmichael, and Sarath K. Guttikunda. "Improving regional air quality predictions in the Indo-Gangetic Plain-Case study of an intensive pollution episode in November 2017." Atmospheric Chemistry and Physics Discussions (2020): 1-29.

Kulkarni, Santosh H., et al. "How Much Does Large-Scale Crop Residue Burning Affect the Air Quality in Delhi?." Environmental Science & Technology 54.8 (2020): 4790-4799.

Pan, X., Ichoku, C., Chin, M., Bian, H., Darmenov, A., Colarco, P., Ellison, L., Kucsera,

T., da Silva, A., and Wang, J.: Six global biomass burning emission datasets: inter-comparison and application in one global aerosol model, Atmospheric Chemistry and Physics, 20, 969-994, 2020.

---

## Author Comment (AC1) · 26 Jan 2021

**Authors' response**

We thank both reviewers for their careful revision and suggestions to further improve the manuscript. Our responses to their comments are given below. Please note that we followed font colour/style as follows:

Black font for Reviewer comments,
Blue font for Authors' response, and
*Blue font* in Italics within quotes for *the modified text in the revised manuscript.*
Note that we also included the page and line numbers in **Blue Bold** font indicating modifications in the revised manuscript.

Thank you,

Corresponding author (on behalf of all co-authors)
* * *
**Reviewer Comments 1 (RC1)**

My main criticisms are to increase the model description, discuss specific anthropogenic sources, and consider particulate air quality.

**Table 3** is included to further include model description

**Page 8 lines 9-10**:

*"Utilizing the emission tracers mentioned above as well as the multiple physics and chemistry options and dynamics schemes (see Table 3),…"*

A substantial portion of the paper is dedicated to evaluating the model skill in simulating CO concentrations. However, the paper does not mention which physics, chemistry, aerosol, and dynamics schemes were used.

Please see previous comment (Table 3 is included). Note that full chemistry-aerosol transport scheme is not applied here, as the study doesn't explicitly deal with aerosols + please see comment below.

A discussion of the gas−phase chemistry mechanism would be especially relevant.

Please note that we have used passive tracer option (not the gas−phase chemistry mechanism) for our purpose. The model setup does not include the deposition and chemical formation of CO from volatile organic compounds (VOCs), which is found to be much smaller, and the deposition processes are minor compared to the CO transport. A discussion of this is already included in the manuscript. Please see Page 8 (third paragraph).

The authors find a larger contribution from anthropogenic emissions than from biomass burning emissions to CO concentrations across the IGP, except within the Punjab for a short period of time during the episode. Examples of these anthropogenic emissions are given (e.g. residential air conditioning systems). However, alternatives to these may be expected to be more likely (e.g. residential solid fuel use for cooking and heating, coal−fired brick kilns). Hence, the chosen examples need explaining.

We agree. Text is modified as:

**Page 18 lines 19-21:**

*"While these anthropogenic urban sources (e.g. road traffic, residential usages such as cooking and heating by solid fuels, industries (including coal-fired kilns) and power plants) ..."*

Air quality in India is mainly important in terms of fine particulate matter (PM2.5) exposure and these biomass burning events also contribute significantly to ambient PM2.5 concentrations (India State-Level Disease Burden Initiative Air Pollution Collaborators 2019, Cusworth et al 2018, Jethva et al 2019). It would be useful to see a discussion of how these episodic emissions contributed to PM2.5 exposures and the associated acute health impacts.

Following Text is included

**Page 11 lines 24-32:**

*"Given that both CO and particulate matter (PM) are usually co-emitted and there exists a reasonably high correlation between them during high pollution episodes, the reported enhanced CO can also be good a indicator of increased PM10 and PM2.5 that are associated with bad air quality and health impacts. As a first-order approximation, high episodic PM estimation can be made using PM/CO linear conversion factors. However, the accurate prediction of particulate matter needs aerosol-chemistry modelling since PM concentration is affected by heterogeneous chemistry and wet/dry removal processes,*

*unlike CO that is mainly affected by atmospheric transport and mixing at regional scales."*

Comments

- 1. Title: The contribution of what? Also, is mesoscale modelling the most accurate term here?

Title is modified to enhance the clarity: *"Using TROPOMI measurements and WRF CO modelling to understand the contribution of meteorology and emissions to an extreme air pollution event in India".*

- 2. This paper has many acronyms. Are all these necessary?

We tried our level best to minimize the usages of the acronyms throughout the manuscript. However, we see that these acronyms are necessary for the completeness and enhancing the readability of the manuscript.

- 3. The paper aims to address 5 questions. It would be useful to have a concise summary of the answers to these questions in the conclusion.

Done. We included following text

**Page 17 line 34 to Page 18 line 15:**

*"Answering to the 5 questions raised in the introduction: 1) the TROPOMI XCO shows a clear enhancement during stubble burning period over the fire emission hotspots but also along the western parts of the IGP, including the national capital of India, Delhi. The detected XCO level over most part of the IGP was about 40 ppm higher than other parts of India. 2) In terms of regional fire CO contribution, in most parts of India, the fire CO emissions peak during the pre-monsoon period (76%) compared to the post-monsoon period (24%). Fire activities over northeast India (NEI) made a significant contribution (57%) to emissions during pre-monsoon months, while the IGP contributed only about 5%. Central (CI) and southern regions (SI) of India add about 33% towards the pre-monsoon fire CO emissions. IGP contributed about 73% of the country's total fire CO emissions during the post-monsoon period. A large amount of fire CO emissions is reported over the state of Punjab within the short period of November 6-9 2018. 3) Comparing model simulations with observations, we find a good agreement between WRF and WFMD with a mean difference of 7 ppb, a standard deviation of 8 ppb, and a spatial correlation coefficient of 0.87. The comparison of WRF CO at surface level with ground level CO measurements from the stations along the IGP for the period of 3-20*

*November 2018 shows a less agreement as compared to the values for XCO, with a correlation coefficient of 0.6 (for the IGP), 0.6 (Delhi) and 0.41 (Punjab). 4) The response of column CO to the surface biomass emission was clearly visible during the enhanced burning period. CO level started to rise over the fire emission hotspots from 6 November and gradually increased during the following days (see Fig. 5). 5) Compared to anthropogenic emission sources, our results imply a minimal role of biomass burning in terms of its contribution to both column and surface enhancements, except for the state of Punjab during the high pollution episodes. This is also consistent with Dekker et al. (2019), which concluded that the low wind speeds and shallow atmospheric boundary layers were the most likely causes for the temporal accumulation and subsequent dispersion of CO during the biomass-burning period in November 2017."*

-4. Page2line10,page3lines3and10,page4lines21and40,page5lines13,14, and 30, page 6 lines 1, 9, 10, and 23, page 7 lines 10, 19, and 28, and page 12 line 34: Define acronyms at first use.

Done. Note that some of the abbreviations are better known than their expanded form in which we followed the ACP guidelines for Abbreviations ( https://www.atmospheric-chemistry-and-physics.net/submission.html#manuscriptcomposition)

- 5. Lines 8−13: Why is the fire−radiative power (FRP) approach more accurate than inventory approaches? In the second half of this sentence, the authors mention inaccuracies in these derived approaches. If by derived the authors imply FRP, then this would be useful to clarify for the reader.

FRP-based approach uses additional observations (fire radiative power) for data-assimilation in addition to emission conversion factors used in inventory-based datasets. Including more fire-related observations is expected to increase the accuracy of derived biomass emissions.

To enhance clarity, we modified sentence as follows:

**Page 4 lines 6-13:**

*"Kaiser et al., (2012) demonstrated an approach for calculating biomass-burning emissions by assimilating satellite-based fire radiative power (FRP) observations. Along with FRP data, this approach derives the combustion rate and trace gas emissions subsequently with land-cover specific conversion factors and emission factors compiled through literature surveys. While the FRP-based approach has clear advantage in enhancing accuracy compared to other inventory/based datasets such as the Global Fire*

*Emission Database (GFED), several studies have indicated inaccuracies in the FRP-derived biomass burning products due to instrument limitations and usage of conversion factors."*

-6. Page4line36,page5line22,page7line27,andpage14lines17and18: Acronyms already defined.

Done

- 7. Page 5 lines 9−19: This is a very long sentence, which requires shortening. Also, this comparison to previous work may be better suited to the Discussion.

Text is modified to shorten its length as follows:

**Page 5 lines 11-16:**

*"An analysis focusing on identifying the sources contributing to the high pollution event in North India during November 2017 using WRF modelling and TROPOMI preliminary operational data was reported in Dekker et al., 2019. Here we present the analysis for the succeeding year, i.e. November 2018. Additionally, this study differs from the previous study as follows: ..."*

Here we mostly compare the approaches and objectives between two studies and state how the present study differs from previous study for giving a general overview. Because of this, we still think that it'd be more suited in the Introduction. However, a discussion of comparison of result and subsequent conclusion are already provided in Sects. 5 and 6.

- 8. Page 5 line 40, and page 6 line 12: Typo: WFM−DOAS.

Done

**Reviewer Comments 2 (RC2)**

In my opinion, the paper is well-written and well-organized. However, I did not find enough evidence for the major conclusions in the paper. I would recommend this paper for publication only after major revision and adding more discussion on the transport of plume in the model and the model performance in capturing these patterns. The uncertainties (either in emission or transport) need to be discussed further.

Thank you for the suggestions, which we tried to include in the revised version of the manuscript.

- Adding backtrajectory analysis can help better understand the transport pattern in the model. - A perturbation run with increased fire emission can help examine your conclusions further and also gives an insight into the transport error of the model. Or using other fire emission inventories - Cross section plots can help with understanding the transport patterns.

We agree that cross-section plots are useful for elucidating the transport patterns involved. We included further analysis using the cross-section plots of the vertical CO distribution. We do not think that the back-trajectory analysis will give additional details about the transport pattern to those already provided by WRF. Please see that the vertical CO distribution (cross-section plots) can clearly show the direction of transport. Note that the uncertainty in the driving meteorology (from WRF or from any other meteorology model) will continue to impact the transport pattern that will be derived from the back-trajectory model.

Perturbed emission runs: We included analysis based on perturbation runs. Biomass burning emissions are perturbed by 50% and quantified how that affects the size of the anomaly seen over the Punjab, Delhi and IGP. Further, we use the emissions from the Global Fire Emissions Database (GFED), version 4 s with small fires (Randerson et al 2012, van der Werf et al 2017) and considered the contribution of small-fires to the fire emission. Accordingly, the impact of including small fires on our simulations over Punjab, Delhi and IGP region is estimated for biomass burning period and for November 2018.

According to the above RC2 comments, major changes done in the manuscript as follows:

**Fig. 10 and 11** (cross-section plots) are included.

**Table S1** is included as supplementary, providing model performance statistics for IGP region during biomass burning period using perturbed emission and small fires.

**Table S2** is included for providing statistics for comparison between WRF and MERRA-2 meteorology for locations in IGP (Agra, Delhi and Barnala).

**Figures S3, S4 and S5** are included, showing the impact of perturbed emissions and small-fires over the IGP region.

**Figure S6** is included for the model meteorology comparison.

Additional changes in the revised manuscript (texts):

**Page 8 lines 11-15:**

*"To assess the impact of small fires on our atmospheric CO mixing ratio simulations, we use another satellite-based fire inventory, the Global Fire Emissions Database version 4s (GFED4s), which includes small fires (Randerson et al., 2012, van der Werf et al., 2017). The dry matter (DM) emissions from GFED4s are converted to CO emissions using emissions factors as given in Akagi et al., 2011."*

**Page 10 lines 18-23:**

*"It should also be noted that very small fires involved can be missed due to MODIS instruments limitations, which may underestimate the fire CO emissions. With a finer spatial resolution of VIIRS (375 m) than MODIS (1 km), VIIRS detected ~20% more active fires at the spatial scale of 0.02° × 0.02° over Punjab and Haryana during the post-monsoon season (Liu et al., 2019)."*

**Page 13 lines 25-33:**

*"A comparison of post-monsoon fire CO emissions over Punjab and Haryana as estimated from five global inventories for the period from 2003 to 2016 indicates the limitation of satellite-derived fire products and the associated uncertainties in the CO fire emissions (Liu et al., 2019)."*

**Page 14 line 24 to Page 15 line 9:**

*"To examine the impact of missed active fires on our WRF results, we perturb GFAS fire emissions by a factor of 50% and quantify how this perturbation affects the size of the anomaly in CO mixing ratios over the IGP region. Note that VIIRS detected ~20% more active fires during the post-monsoon season over Punjab and Haryana. Using the perturbed GFAS emissions, we estimate the relative increase in modelled XCO contribution arises from increased biomass emissions. With the increased emissions, we see an increment of XCO contribution ranging approximately from 5 to 25 ppb during biomass burning period over IGP region, mostly over Punjab, Haryana and Delhi (see Fig(s). S3 and S5). As for the model-observation performance statistics, a slight improvement is found for XCO over IGP region with this perturbed simulation (see supplementary Table S1)."*

*"For allocating small fires over the model domain, we use GFED4s fire product including fire fractions stemmed from the small fire burned area. The small fire boost in GFEDv4s is calculated based on active fire hotspots and burned area observations from MODIS surface reflectance (Randerson et al., 2012). The difference in fire emission fields in GFED4s relative to GFAS is derived over the model domain and is applied to WRF for quantifying the fire CO contribution that also includes small-fires. While*

*including small-fires based on GFED4s has improved the model-observation mean bias over IGP region for surface CO mixing ratio during biomass burning period, we see a minimal improvement for XCO (see supplementary Table S1). Enhancing the fire emission by incorporating small-fires resulted in overall increment of XCO concentration ranging from 20 to 40 ppb, however most of the contributions arise from small-fires are seen only over Punjab and some parts of Haryana (see Fig(s). S3 and S5). Based on GFED4s, we quantify the effect of small-fires on the modeled atmospheric CO plumes. The addition of small fires contributed to an increment of 12.2 % surface CO over Punjab and Haryana, and the small-fire contribution is reduced to 8.6 % and 4.3 % over Delhi and IGP respectively. In case of XCO, there exists only a minimal impact of small fires on mixing ratios, which are estimated to be 2.5 % over Punjab and Haryana, 1.4 % over IGP, and 0.8 % over Delhi. The difference in the contribution of small-fires between surface CO and XCO can be explained by the meteorology conditions prevailed (see Sect. 5.5)."*

**Page 15 lines 27-35:**

*"For further analysing the effect of meteorological conditions, we use WRF-simulated meteorology due to the lack of observations of wind and PBL height in this region. An inter-model comparison of WRF meterology with corresponding variables from reanalysis data provided by Modern-Era Retrospective Analysis for Research and Applications, Version 2 (MERRA-2) is performed to assess the overall agreement (see Table S2 and Fig. S6). Note that MERRA-2 is an assimilation product at an approximate spatial resolution of 0.5° × 0.625°, publicly available online through* https://gmao.gsfc.nasa.gov/reanalysis/MERRA-2/. *More information on MERRA-2 and the assimilation system can be seen in Gelaro et al. (2017)."*

**Page 16 lines 10-18:**

*"Figures 10 and 11 provide transport patterns involving the vertical distribution of CO biomass burning contribution and total CO mixing ratio respectively during biomass burning period. Vertical cross-sections show an impact of fire emission over Delhi during biomass burning period (40 to 120 ppb), peaking its boundary layer CO contribution (> 110 ppb) on November 7 (Fig. 10). On the other hand, the total CO shows peak values (> 550 ppb) on November 9, indicating a significant additive contribution from anthropogenic fluxes in addition to biomass burning together with winter meteorology conditions prevailed over the region (see Fig. 11). A consistently low PBL height can be clearly seen during these days, which traps CO plumes in the lower boundary layer due to less extent of vertical mixing."*

General comments: - Add more details of WRF-Chem model configuration. Any nudging or re-initialization of the model? By running the model freely for a month will increase the errors in the model.

Done. Please see our response to RC1. Also we included the text as follows:

**Page 7 lines 18-19:**

*"The model is reinitialized each day with ERA5 meteorology and allowed for 6 h spin-up time."*

- There is no model performance evaluation with respect to meteorology variables. Please consider adding it to the discussion.

We agree that a model performance evaluation with relevant and continuous observed meteorological variables would add the confidence level of our model. However, there is a general lack of such detailed observations in this region of interest, which limits this performance evaluation. Nevertheless, note that we see a good agreement with column CO that is also controlled by atmospheric transport patterns. To assess the overall agreement in meteorology variables, we included an inter-model comparison of WRF meteorology with corresponding variables from reanalysis data provided by MERRA-2 (see Table S2 and Fig. S6). Also see our previous comments regarding meteorology validation.

We added the following text:

**Page 15 lines 27-29:**

*"For further analysing the effect of meteorological conditions, we use WRF-simulated meteorology due to the lack of observations of wind and PBL height in this region."*

Also we added:

**Page 12 lines 25-27:**

*"the results indicate shortcomings in the model that can be refined by better representation of atmospheric transport (including model initialisation) and emission."*

- Even after reading P5 L11-19, I still think that this study is very similar to Dekker et.al 2019. Consider adding more analysis to your study.

We disagree with this. Compared to Dekker et.al 2019, the present involves

1) Different Period of analysis
2) Completely different WRF-model set up: model domain, vertical resolution, different emission inventory, different physics/dynamics schemes
3) Usage of different satellite algorithm product for the analysis
4) Detailed analysis of burning period and non-burning period using column observations and flux data.

- Consider adding evaluation of fire emission performance. Perhaps compare the emission with VIIRS active fire points to see how emission inventory underestimated CO emission. What about other fire emission inventories?

We agree that there can be underestimation of CO emission in the inventory used since very small fires involved can be easily missed due to MODIS instruments limitations. By comparing TROPOMI observations with our model simulations, we do not, however, think that there is a substantial underestimation of biomass emissions in this case unless the EDGAR CO is largely overestimated. We acknowledge that VIIRS is more small-fire optimized that MODIS instruments. At the same time, please note that it is not a straight forward to compare VIIRS fire product with existing emission inventory for evaluating the CO fire emissions. However, we use the details that VIIRS detected ~20% more active fires during post-monsoon season over Punjab and Haryana. To accommodate this, we perturb the fire emission by a factor of 50 % and included the analysis in the revised manuscript. Please see our previous comments regarding perturbed emission **(Page 14 line 24 to Page 15 line 9)** and text indicating the MODIS limitation **(Page 10 line 18-23).**

Also please note that the following sentence (about the limitation) is already there in the manuscript: **(Page 13 lines 19-22)** *"The GFAS fire emissions are partly based on the MODIS satellite instrument, and the limited resolution of the instrument misses many small fires, including biomass burning over India (Cusworth et al., 2018)."*

- Have you looked at the performance EDGAR inventory in this region in the literature? Can errors in EDGAR partly justify the large errors in surface CO?

Yes, we have already indicated this possibility of errors in surface fluxes that impact the surface CO simulations. To make it clearer, the sentence is modified as follows:

**Page 13 lines 27-30:**

*"however there is a non-trivial mean bias which can be attributed to issues with simulating transport (including the emission release height) and PBL dynamics in WRF*

*as well as the variability in emission fluxes **(both EDGAR and GFAS**) which is likely to be not sufficiently well represented in the emission inventories used."*

- Map of the contribution of emission CO source to total in November 2018 can be helpful.

The figure is included as supplement **(Fig(s) S1 and S2).**

**Page 13 lines 38-40:**

*"The relative contributions of different emission sources and processes to the WRF CO column, as summarized in Table 4, clearly indicate the dominance of anthropogenic signals over biomass burning signals on the XCO enhancements (see supplementary figures S1 and S2)."*

Specific Comments

P5 L15 – I did not find (2) in the paper. Where did you examine the regional distribution of CO for the entire year? You also did not specify which model configuration you used and why you used these options.

Please see Sect. 5.1 and Fig.2. Also see our response to RC1 for model configuration and concise summary of results in the conclusion section.

P6 L40 – I recommend using a map marking station locations rather than listing stations in Table 2. Perhaps replacing Table 2 with Figure 8.

Figure 8 is the map marking station locations. We would like to keep Table 2 for providing location details of stations more clearly as some of the station details are not very clear in the map due to overlapping.

P7 L2-9 – Add more details about the WRF-Chem configuration and options.

See our response to RC1 for model configuration.

P7 L34 – consider adding Pan et al 2020 reference.

Done

P7 L37 – So all emissions from fires are released from the surface? How does this limit the model performance?

Releasing emissions from the surface would have only minor effect on the model performance in simulating column integrated CO. It may have some effect on surface CO simulations. We added the following sentence:

Page 13 lines 27-30

*"..a non-trivial mean bias which can be attributed to issues with simulating transport (including the emission release height) and PBL dynamics in WRF as well as the variability in emission fluxes (both EDGAR and GFAS) which is likely to be not sufficiently well represented in the emission inventories used."*

P8 L1 – Multiple physics and chemistry options and dynamics schemes are not discussed in the paper.

See our response to RC1 for model configuration

P9 L22-26 – consider referencing Kulkarni et al. 2020 here on fire activities

Done. Following text is included

Page 10 lines 7-9:

*"This is also consistent with the distribution of total fire counts over IGP region during the post-monsoon period as seen in Kulkarni et al. (2020)."*

P10, sec 5.3.1. – It is difficult to see the differences between model and obs in Figure 3. Please add bias map for Nov 6-9 (similar to Fig 4 a). Perhaps you may want to reorganize the panels in fig 3 and 4. Consider using different color bars for bias maps and XCO maps.

Included **Fig. 4c and 4d**. Also we used diverging color map for making it clearer.

P11 L4-7 – You have a relatively large domain and you are focusing in a smaller region in the domain with high biomass burning activities. Averaging the biases over the whole domain cannot really help with your discussion. I suggest providing statistics only for IGP region.

We disagree. This part deals with the evaluation of WRF modeling with TROPOMI observations for the whole Indian region. It is not only dealing with biomass burning activities. During biomass burning period, IGP region is focused for the stats (see next paragraph). The revised manuscript includes the statistics restricting only for IGP region

during the biomass-burning period (please see **Table S1**).

P11 L8 – by "biomass-burning period" do you mean the month of November or just Nov 6-9? Fig 4 only shows November.

Biomass burning period is Nov 6-9, 2018. The details of Fig.4 provided in the ACPD version of the manuscript was wrong and is corrected now (we removed "Fig. 4 demonstrates these differences.")

Figure 5 – add another column with CO emission from fire similar to Fig 2-b but for each day. Discuss how fire emission varied day by day and if model captured it.

Please note that Fig.5 is about mixing ratios, discussion is not about how to capture GFAS emissions which is already used as emission in the model. Please note the inclusion of cross-section plots. Also please see our previous comments regarding transport patterns.

P11 L17 – Looking at Fig 5 it looks like on Nov 6 and 7 model didn't capture emissions correctly and on Nov 8 model didn't transport the pollutant correctly. Please discuss the errors further. There are other papers that looked at a similar problem for Nov 2018 such as Kumar et al., 2020 and Roozitalab et al., 2020, please compare your findings with their conclusions.

We agree that the model has uncertainty, and is discussed in Sect. 5.3.1. But it is encouraged to look at the fact that both TROPOMI observations and simulations suggest a southeast transport of plume that increases the CO concentration over Delhi and Agra during 8 and 9 November. We cannot directly compare our CO results with suggested references as those studies are for the year 2017 and for AOD/PM2.5. However we noted that Roozitalab et al., 2020 reports an underestimation of the WRF-Chem simulated AOD during an intensive fire period for IGP except Punjab, though used entirely different model set up compared to this study. Also we note that Kumar et al., 2020 found significant inconsistencies in the model.

We included a discussion as follows:

**Page 12 lines 16-27:**

*"Based on VIIRS AOD and WRF-Chem simulations using different chemical and meteorology boundary conditions and biomass burning emissions, Roozitalab et al., 2020 assessed the model performance over the IGP region during an intensive fire period in November 2017 and reported an underestimation of AODs for the entire IGP region,*

*except for Punjab. Further, Kumar et al., 2020 found a considerable impact of uncertainties in the WRF meteorology on simulated PM2.5 concentrations over IGP during crop residue burning period in November 2017. Note that our study also reports slight underestimation of WRF compared to TROPOMI CO observations over IGP during biomass burning period. Though we cannot directly compare AOD/PM2.5 results during November 2017 from previous studies with our CO simulations during November 2018, the results indicate shortcomings in the model that can be refined by better representation of atmospheric transport (including model initialisation) and emission."*

P11 L 18 – Fig 6 Can you mark these locations on the map on Fig 5? Please use same XCO range for all three sub plots.

Done

P11 L18-34 – Adding backtrajcetory analysis can be beneficial for the argument here. More discussion on model errors in capturing transport pattern is needed. Before doing inverse modeling to constrain emission you need to understand the model errors.

See our previous comments. Note that cross-section plots are included.

P11- L35 – Are Punjab and Delhi region showed in Fig 6 and 7 the same regions? I suggest marking these regions on the map for both column and surface observations.

Please note that Fig.6 is for column observations and Fig. 7 is for surface measurements from available sites. Region selection is identical, but there are spatial gaps in the surface measurements. Fig.8 has indicated the locations. Also see our statement *"The data are averaged in a 100 km × 100 km square around the centre of each city."*

P12 L3 – What do you mean by larger variability? To me, it looks like obs CO variability is lower in Punjab compared to Delhi

Corrected.

**Page 12 line 1:**

*".. larger variability associated with biomass emissions compared to other stations."*

P12 L28 – From Fig 5 it looks like model did not have any fires near Punjab on Nov 6 and Nov 7. You can add a similar discussion in this section by referring to fig 5. Also adding backtrajectory analysis can help with understanding the transport patterns.

See our comments above for back-trajectory. Note that Fig. 5 is for XCO. Please see Fig. 7 for this. Also see Fig. 10 in which we see contribution of fires on November 7. Further, see our previous comments about the vertical cross-section plots.

P13 L10 – What do you mean by observed variability? The influence of background is not minimal when looking at column CO.

Please note that this part refers to Fig. 7 and surface CO, not the column CO.

To make it clearer, we modified the text as:

Page 14 lines 14-15:

*"..background CO concentrations to the surface level CO observed variability is minimal"*

P13 L20 – Since GFAS underestimates fire emissions do you expect a higher contribution of fire CO to total CO in Punjab in reality?

By looking at the TROPOMI observations and comparing it with WRF over Punjab region for biomass burning peak days, we assess that a part of the model-observation mismatch can be attributed to the underestimation of modelled fire CO contribution.

We included the text as follows:

Page 15 lines 16-19:

*"This is also supported by our study in which we find underestimation of the total CO concentration in Punjab during biomass burning period and a part of this model-observation mismatch can be attributed to the underestimation of modelled fire CO contribution."*

P13 L32 – Please overlay obs surface CO and model in these plots. Also, use the same range for Y axis for all plots. Comparing with observed meteorology variables can greatly benefit the discussion here and help justify the large biases of surface CO for some days.

We used different ranges for better visualization of signals and their variations. Fig.7 and Fig.9 are separated in order to avoid too many details in one figure, making it messy. See comments above for observed meteorology variables.

We included the following text in Fig.7 caption

*"Note that different Y-axis scale ranges are used in panels for better visualization of signals."*

P14 L1 – You got to this conclusion based on correlation numbers from Nov 3-20 and I don't think that is enough. How are the correlations during Nov 6-9?

A statement with the correlations is added in the text. Also please see Table S1 for the statistics during biomass-burning period.

We modified the text as:

*"Similar correlations of CO with modelled PBLH (-0.87 (IGP), -0.83 (Delhi), -0.65 (Punjab)) and wind speed (-0.46 (IGP), -0.70 (Delhi), -0.26 (Punjab) exist for the biomass burning time period Nov 6-9, 2018."*

P14 L15 – The level of contribution of meteorology to regional air quality can vary day by day. Have you looked at other studies that looked at Nov 2018 such as Kumar et al., 2020 and Roozitalab et al., 2020?

Yes, we do agree that the meteorological contribution can vary on daily basis. This is the reason why we simulated meteorology and take into account hourly variations of CO resulting from meteorology as well (in addition to emissions). Also note that those studies referred are for November 2017, not for November 2018 (our study). Please see our comments above with respect to these references and modifications in the revised manuscript.

Table 3. Overview of WRF-Chem model set-up.

| | | |
|---|---|---|
| | Configuration | Single domain with horizontal resolution of 10 km having 307 × 407 grid points and 39 vertical levels. |
| | Vertical coordinates | Terrain-following hydrostatic pressure vertical coordinates |
| Domain | Basic Equations | non-hydrostatic |
| | Time integration | 3rd order Runge-Kutta split-explicit |
| | Time-step | 60 s |
| | Spatial integration | 3rd and 5th order differencing for vertical and horizontal advection respectively |
| | Radiation | Rapid Radiative Transfer Model (RRTM) for long-wave & Dudhia for short-wave |
| | Microphysics | WSM 3-classic simple ice scheme |
| Physics/Dynamics Schemes | PBL | YSU |
| | Surface layer | Monin-Obukhov |
| | Land-surface | NOAH LSM |
| | Cumulus | Grell-Devenyi ensemble scheme |
| | Chemical mechanism | Greenhouse gas tracer option (passive tracer) using previous simulations |

| Chemistry Options | | to initialize tracer fields. |
| --- | --- | --- |
| | Emission input and specification | Setting (=16) for fluxes and emissions to passive tracers. |

[Figure]

**Figure 10.** Vertical cross section of CO mixing ratio that arises from biomass burning emissions during 6-9 November 2018. Cross-sections are over Delhi for 1:30 PM (local time). The black arrow indicates the location of Delhi.

[Figure]

**Figure 11.** Same as Fig. 10, but showing total CO distribution.

Table S1. Model performance statistics for IGP region during biomass burning period. WRF represents WRF-CO simulations as described in Sect. 3. WRF (Exp. 1) represents the WRF CO simulations by perturbing GFAS biomass burning emissions by an increment of 50%. WRF (Exp. 2) represents WRF-CO simulations by including the contribution from small fires based on GFED4s.

| Period | Model runs | Column (XCO) | | | Surface (CO) | | |
|---|---|---|---|---|---|---|---|
| | | R | MB (ppb) | SD (ppb) | R | MB (ppb) | SD (ppb) |
| November 2018 | WRF | 0.77 | 6.6 | 7.2 | 0.60 | -161 | 295 |
| | WRF (Exp. 1) | 0.77 | 8.5 | 7.0 | 0.55 | -145 | 297 |
| | WRF (Exp. 2) | 0.52 | 10.0 | 14.0 | 0.55 | -120 | 301 |
| 6 – 9 November 2018 | WRF | 0.57 | -2.7 | 15.0 | 0.74 | -117 | 217 |
| | WRF (Exp. 1) | 0.62 | 0.3 | 14.0 | 0.72 | -76 | 226 |
| | WRF (Exp. 2) | 0.51 | 1.9 | 22.0 | 0.7 | -2 | 246 |

Table S2. Inter-model comparison between WRF and MERRA-2 meteorology for locations in IGP (Agra, Delhi and Barnala)

| Station | Temperature | | | u-component | | | v-component | | |
|---|---|---|---|---|---|---|---|---|---|
| | R | MB ($^o$C) | SD ($^o$C) | R | MB (m s$^{-1}$) | SD (m s$^{-1}$) | R | MB (m s$^{-1}$) | SD (m s$^{-1}$) |
| Agra | 0.87 | 0.29 | 0.87 | 0.81 | 0.75 | 0.88 | 0.70 | -0.54 | 1.10 |
| Delhi | 0.82 | 0.44 | 0.98 | 0.74 | -1.02 | 1.28 | 0.83 | 0.72 | 1.01 |
| Barnala | 0.75 | -0.14 | 0.96 | 0.53 | -0.93 | 1.62 | 0.67 | -0.39 | 1.22 |

[Figure]

**Figure S1.** Contribution from emission CO sources to the total XCO in November 2018: (a) anthropogenic contribution (b) background contribution (c) biomass burning contribution. The contributions are derived using respective WRF tracers (see Sect. 5.4)

[Figure]

**Figure S2.** Same as S1, but restricting the period to 6-9 November, 2018 (biomass burning period)

[Figure]

**Figure S3. (a)** Differences of CO total column mixing ratios (WRF (Exp. 1) − TROPOMI/WFMD) averaged over the month of November 2018. **(b)** Histogram of the differences. (c) Same as (a), but restricting the period to 6-9 November 2018. (d) Same as (b), but restricting the period to 6-9 November 2018. WRF (Exp. 1) represents the WRF CO simulations by perturbing GFAS biomass burning emissions by an increment of 50%

[Figure]

**Figure S4.** Same as Fig. S3, but using WRF (Exp. 2) simulations. WRF (Exp. 2) represents WRF-CO simulations by including the contribution from small fires based on GFED4s.

[Figure]

**Figure S5. (a)** Differences of CO total column mixing ratios (WRF (Exp. 1) −WRF) averaged over the month of November 2018. **(b)** Differences of CO total column mixing ratios (WRF (Exp. 2) −WRF) averaged over the month of November 2018. (c) Same as (a), but restricting the period to 6-9 November 2018. (d) Same as (b), but restricting the period to 6-9 November 2018. WRF represents WRF-CO simulations as described in Sect. 3. WRF (Exp. 1) represents the WRF CO simulations by perturbing GFAS biomass burning emissions by an increment of 50%. WRF (Exp. 2) represents WRF-CO simulations by including the contribution from small fires based on GFED4s.

[Figure]

**Figure S6.** Inter-model comparison between WRF and MERRA-2 meteorological variables for November 2018. (a) Temperature at 2 m (b) u-component at 10m and (c) v-component at 10m